# HLA allele-calling using multi-ancestry whole-exome sequencing from the UK Biobank identifies 129 novel associations in 11 autoimmune diseases

Guillaume Butler-Laporte [1,2,3✉], Joseph Farjoun[2], Tomoko Nakanishi [2,4,5,6], Tianyuan Lu [2,7], Erik Abner [8], Yiheng Chen[2], Michael Hultström [1,2,9,10], Andres Metspalu [8], Lili Milani [8], Reedik Mägi [8], Mari Nelis[8], Georgi Hudjashov[8], Estonian Biobank Research Team*, Satoshi Yoshiji[2,4,5,6], Yann Ilboudo[2], Kevin Y. H. Liang[2], Chen-Yang Su [2], Julian D. S. Willett [2], Tõnu Esko[8], Sirui Zhou[5], Vincenzo Forgetta[2,7], Daniel Taliun[5] & J. Brent Richards [1,2,5,7,11,12]

The human leukocyte antigen (HLA) region on chromosome 6 is strongly associated with many immune-mediated and infection-related diseases. Due to its highly polymorphic nature and complex linkage disequilibrium patterns, traditional genetic association studies of single nucleotide polymorphisms do not perform well in this region. Instead, the field has adopted the assessment of the association of HLA alleles (i.e., entire HLA gene haplotypes) with disease. Often based on genotyping arrays, these association studies impute HLA alleles, decreasing accuracy and thus statistical power for rare alleles and in non-European ancestries. Here, we use whole-exome sequencing (WES) from 454,824 UK Biobank (UKB) participants to directly call HLA alleles using the HLA-HD algorithm. We show this method is more accurate than imputing HLA alleles and harness the improved statistical power to identify 360 associations for 11 auto-immune phenotypes (at least 129 likely novel), leading to better insights into the specific coding polymorphisms that underlie these diseases. We show that HLA alleles with synonymous variants, often overlooked in HLA studies, can significantly influence these phenotypes. Lastly, we show that HLA sequencing may improve polygenic risk scores accuracy across ancestries. These findings allow better characterization of the role of the HLA region in human disease.

[1] Department of Epidemiology, Biostatistics and Occupational Health, McGill University, Montréal, QC, Canada. [2] Lady Davis Institute, Jewish General Hospital, McGill University, Montréal, QC, Canada. [3] Wellcome Trust Centre for Human Genetics, University of Oxford, Oxford, UK. [4] Kyoto-McGill International Collaborative School in Genomic Medicine, Graduate School of Medicine, Kyoto University, Kyoto, Japan. [5] Department of Human Genetics, McGill University, Montréal, QC, Canada. [6] Research Fellow, Japan Society for the Promotion of Science, Tokyo, Japan. [7] 5 Prime Sciences Inc, Montreal, Quebec, Canada. [8] Estonian Genome Center, Institute of Genomics, University of Tartu, Tartu, Estonia. [9] Integrative Physiology, Department of Medical Cell Biology, Uppsala University, Uppsala, Sweden. [10] Anaesthesiology and Intensive Care Medicine, Department of Surgical Sciences, Uppsala University, Uppsala, Sweden. [11] Department of Twin Research, King's College London, London, UK. [12] Infectious Diseases and Immunity in Global Health Program, Research Institute of the McGill University Health Centre, Montréal, Québec, Canada. *A list of authors and their affiliations appears at the end of the paper.
✉email: guillaume.butler-laporte@mcgill.ca

The HLA[1] gene complex is a highly polymorphic region of the human genome with a striking linkage disequilibrium (LD) pattern. While genetic variants in HLA are often strongly associated with multiple auto-immune and infectious diseases[2,3], genome-wide association studies (GWAS) cannot easily be fine-mapped to likely causal variants, and consequently specialized methods are required to improve statistical power and fine-mapping[3]. Hence, in most current genetic association studies of HLA, the unit of variation is not usually a single nucleotide polymorphism but rather a whole HLA gene version or haplotype, known as an HLA allele.

By convention, HLA allele names start with the gene name, followed by up to four sets of digits (also called fields 1 to 4), each separated by a colon. From left to right, these digits provide information on the allele's serological specificity, HLA protein, synonymous variants, and non-coding (i.e., intronic) variants. For example, HLA-A*01:01:01:01 is one such allele for gene HLA-A. The use of HLA alleles in association tests, known as HLA fine-mapping, has higher statistical power than single nucleotide polymorphism-based approaches and allows for a better understanding of the role of the HLA region in a wide range of conditions[2,4–7]. It can help identify targets for novel medicines and improve our ability to identify populations at risk for immune- and infection-mediated disease.

For the HLA fine-mapping to be possible, HLA alleles must be accurately assigned to study participants. The most common and cost-effective method is to impute HLA alleles from variants typed with genotyping arrays[8–13]. However, the imputation requires large and diverse HLA reference panels, access to which still needs to be improved, and is less accurate for individuals of underrepresented ancestries[14] and rarer alleles[15]. Using sequencing data to *call* HLA alleles eliminates the need for such a reference panel and may provide better accuracy of individual-level HLA alleles, resulting in improved fine-mapping and statistical power.

In this study, we used the UKB[16] release of 454,824 WES sequences[17] to call each participant's HLA alleles using the HLA-HD algorithm[18]. HLA-HD provides reliable HLA allele calling from short-read sequencing[19] and is easily scalable on a cloud computing environment like DNAnexus (Palo Alto, California, USA). It is also the only published HLA allele calling algorithm that provides 3-field calls and whose IMGT-HLA internal reference can be updated to use the most recent one. We then provide a comprehensive report on the HLA allele landscape in UKB participants of 5 ancestries (African [AFR], Admixed American [AMR], East Asian [EAS], European [EUR], South Asian [SAS]), and we compare our results to imputed HLA alleles currently available in UKB participants. We assessed the improvement in statistical power by performing HLA allele and amino acid association studies on 11 auto-immune traits across all genetic ancestries. Lastly, we built polygenic risk scores (PRS) incorporating HLA alleles for these traits. Our findings should allow for a better understanding of the role of HLA alleles in disease and better risk stratification.

## Results

**HLA allele calling from WES**. HLA-HD was used to call HLA alleles for 454,824 participants at 3-field resolution (representing the allele's serological specificity, HLA protein, and synonymous variants). We used the UKB whole-genome genotyping (unavailable in 1283 participants) projected on the 1000 Genome reference to estimate genetic ancestry. We found that this cohort included 8725 participants of AFR genetic ancestry, 2898 of AMR genetic ancestry, 2647 of EAS genetic ancestry, 429,822 of EUR genetic ancestry, and 9449 of SAS genetic ancestry (see Methods).

The UKB WES target regions provided reads at 31 HLA genes. These included 12 HLA Class I genes (6 protein-coding genes) and 19 HLA Class II genes (13 protein-coding). Class I genes were generally well covered (Supplementary Data 1 and Supplementary Fig. 1), with all participants having more than 20 reads aligning to HLA-A, HLA-C, HLA-E, HLA-F, and HLA-G, while only 109 of 454,814 (0.02%) participants had less than 20 reads at their HLA-B gene. Class II genes were also well covered, except for HLA-DQA1, which was less well-covered; while 95.9% of participants had calls at HLA-DQA1, 34.4% of them had less than 20 reads at exon 2. While it is challenging to assess read coverage for HLA-DRB3 to HLA-DRB9 given that these genes are not carried by every individual, HLA-DRB1 was well covered (0.23% of participants with less than 20 reads). All Class I genes were found in each ancestry. However, the EUR cohort was the only one for which all Class II genes were found, with HLA-DPA2 and HLA-DRB9 absent in all other ancestral cohorts and HLA-DRB6 also missing in the AFR, AMR, and EAS cohorts.

As expected, the number of unique HLA alleles was the highest in the larger EUR cohort, at 5,295, and ranged from 985 (EAS) to 1527 (SAS) in smaller cohorts of the other four ancestries (Fig. 1a). When adjusted by sample size, the AMR and EAS genetic ancestry participants had the largest number of alleles (0.432 and 0.372 alleles per participant, respectively), while the EUR cohort had the lowest (0.012) (Fig. 1b). The fact that this ratio is the smallest in the EUR cohort is likely due to fact that past a certain sample size, the likelihood of finding a new additional allele will decrease. The finding that the AMR participants have a larger number of HLA alleles is consistent with other studies and reference panels which showed that native American populations have a high number of HLA alleles absent in other populations[20–22]. As expected, most of the HLA alleles were rare (minor allele frequency [MAF] < 1%) in all ancestries, with 166 out of 5295 alleles with MAF > 1% in the EUR cohort and 209 out of 1304 alleles with MAF > 1% in the AMR cohort (Supplementary Fig. 2). Similarly, the first 25 most common alleles in each ancestry account for > 90% of total variation frequency. (Fig. 1c). Lastly, HLA Class I genes (including pseudogenes) showed the highest diversity, with an average of 586.0 alleles per gene compared to 194.4 for HLA Class II genes. The highest number of alleles was found in the EUR cohort, and the lowest in the EAS cohort (Fig. 1d), but once again, the observation was mirrored when accounting for sample size (Fig. 1e). Lastly, to adjust for the large differences in participants of each ancestry which could saturate the number of alleles found in each group, and hence bias comparisons on number of alleles per ancestry groups, we down sampled 10,000 times each group to have the same sample size and number of HLA alleles as the smallest group (EAS). This analysis was only done for classical HLA genes, as they had enough unique alleles for the simulations to be stable. As expected, the AMR ancestry cohort had more expected unique alleles than other groups in all genes, except for HLA-DPA1 where the AFR cohort, had more (Supplementary Data 2 and Supplementary Fig. 3). A complete list of alleles and their frequencies at 3-field and 2-field resolution is available in Supplementary Data 3 and Supplementary Data 4.

**Comparison to imputed alleles**. The UKB provides HLA allele imputation using the HLA:IMP*2 software[13] for 11 genes at 2-field resolution: HLA-A, HLA-B, HLA-C, HLA-DQA1, HLA-DQB1, HLA-DPA1, HLA-DPB1, HLA-DRB1, HLA-DRB3, HLA-DRB4, HLA-DRB5. These HLA imputation results have been used extensively in the literature in the past. While they do not represent a perfect gold standard (especially in non-European populations), they still provide an important resource to compare

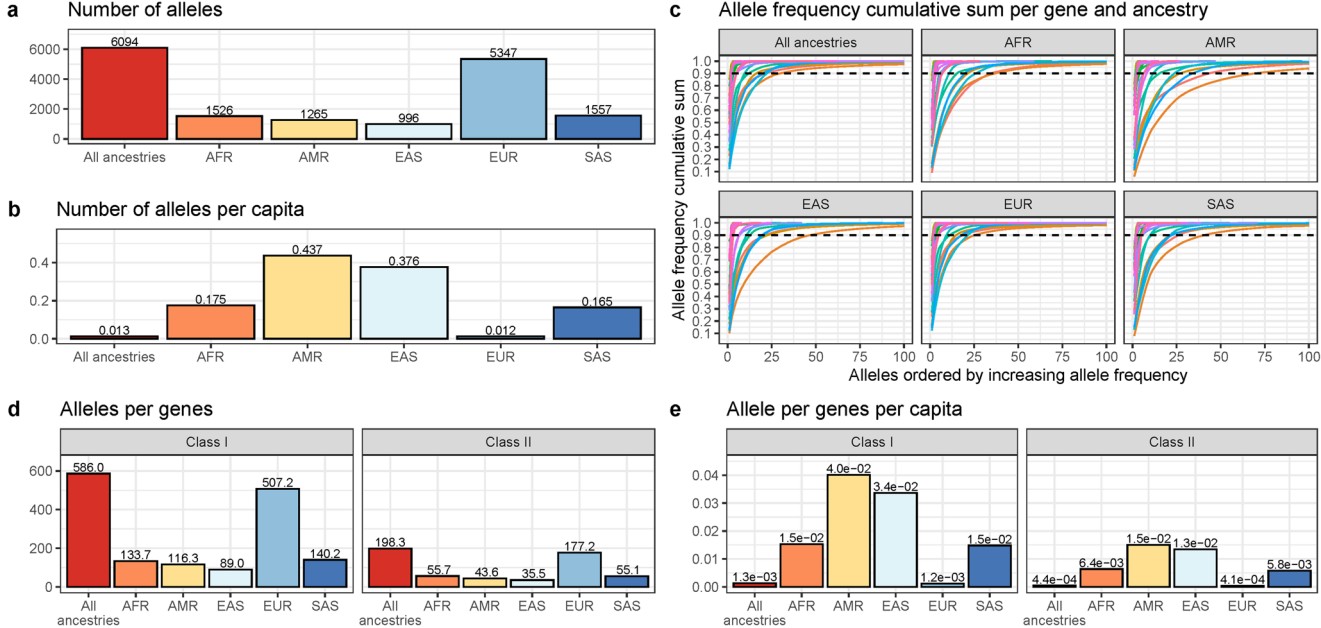

**Fig. 1 Summary of HLA alleles and their distribution per genetic ancestry. a** Number of 3-field HLA alleles per continental genetic ancestry. **b** Number of 3-field HLA alleles per genetic ancestry, divided by the number of participants in each ancestry. **c** Cumulative 3-field allele frequency. Each line represents a different HLA gene. Dashed line at 90%. Note that some genes are not present in all participants (e.g., *HLA-DRB3*), and their true cumulative allele frequency sum would be less than 100%. Hence, frequencies are given as allele count for allele divided by total number of alleles at that gene, then added cumulatively starting with the alleles with the highest frequencies. **d** The average number of alleles per gene stratified by HLA class. **e** The average number of alleles per gene divided by the number of participants in each cohort stratified HLA class. All analyses in this figure were limited to the protein-coding genes.

our HLA calls against. We, therefore, compared concordance between the sequenced alleles at 2-field resolution and the previously imputed UKB alleles. It is crucial to note that there is currently no perfect reference standard against which to compare HLA calling or imputation methods, and that any comparison will suffer from considerable ambiguity.

Nevertheless, we set out to compare imputed and called alleles in the UKB. The complete set of imputed alleles included 196 in Class I and 136 in Class II genes. 10 out of these 332 alleles were absent among the sequenced HLA alleles. However, all 10 of these alleles had low frequency (MAF < 0.04%), suggesting that these may have been imputation mistakes because the imputation accuracy decreases with the allele frequency.

While allele concordance was excellent for HLA Class I genes (90.4% for *HLA-A*, 89.5% for *HLA-B*, and 91.1% for *HLA-C*) and *HLA-DPA1* (96.8% for *HLA-DPA1*), it dropped substantially for other genes (as low as 47.2% for *HLA-DQA1*) (Supplementary Data 6–11). However, this concordance calculation did not account for the considerable increase in the number of discovered HLA alleles in recent years, which has more than doubled since the original publication of the HLA:IMP*2 software[13]. Specifically, HLA-HD had more alleles at its disposal that it could assign to each participant, this includes novel alleles that were split form old ones and that are not included in the HLA:IMP*2 algorithm. This therefore clearly biases any direct comparison against HLA-HD, as any call using a more modern HLA allele would be considered automatically wrong by HLA:IMP*2. Hence, we also looked at an adjusted concordance rate by only looking at the concordance of called alleles that were present in the HLA:IMP*2 database. As expected, the adjusted concordance was much greater, with only two genes scoring less than 90%: *HLA-DQA1* and *HLA-DRB4* (Fig. 2 and Supplementary Data 6–11). For the latter two genes, this was due HLA alleles that had not been imputed but were called in more than 95% of cases of

discordance. This further highlights the increased power of HLA sequencing over imputation. For *HLA-DQA1*, we suspect that this is consistent with it having been the most poorly sequenced genes, as discussed above (Supplementary Fig. 1).

Of the participants whose alleles did not completely match, the mismatch was primarily due to HLA alleles of low allele frequencies (MAF < 1%). The mean allele frequencies of those participants ranged from 1.16% for *HLA-DPA1* to 0.09% for *HLA-B* (Supplementary Data 5). For all genes and ancestries, lower allele frequency was associated with a lower concordance rate (Supplementary Fig. 4). Moreover, there was a significant decrease in both allele concordance and adjusted concordance in participants of non-EUR ancestries (Fig. 2 and Supplementary Data 6–11), for both classes of HLA genes. While the average allele concordance and adjusted concordance for HLA Class I genes in the EUR cohort was 91.0% and 96.9%, they dropped to 75.5% and 88.4% in AFR, to 81.1% and 89.9% in AMR, to 78.7% and 86.8% in EAS, and to 81.2% and 90.1% in SAS. Likewise, EUR participants' mean concordance and adjusted concordance for HLA Class II alleles was 73.9% and 91.5%, which decreased to 68.5% and 82.4% in AFR, to 70.4% and 83.3% in AMR, to 69.5% and 82.9% in EAS, and to 70.5% and 81.2%in SAS. Finally, using different allele dosing QC threshold for the imputed alleles only had a mild effect on concordance results, with an average decrease in adjusted concordance of 0.93 percentage point (range: 0.10–3.0) when using liberal dosage thresholds (see Methods), and an average increase in adjusted concordance of 2.01 percentage point (range 0.2–6.2) when using a stricter threshold (see Supplementary Data 6–11 for full comparisons).

Hence, HLA sequencing improves accuracy compared to previously imputed alleles, this improvement in not fully explained by an increase in the number of known HLA alleles, and, as suspected, the non-EUR genetic ancestry individuals and those who carry rarer alleles suffer the most from the decrease in

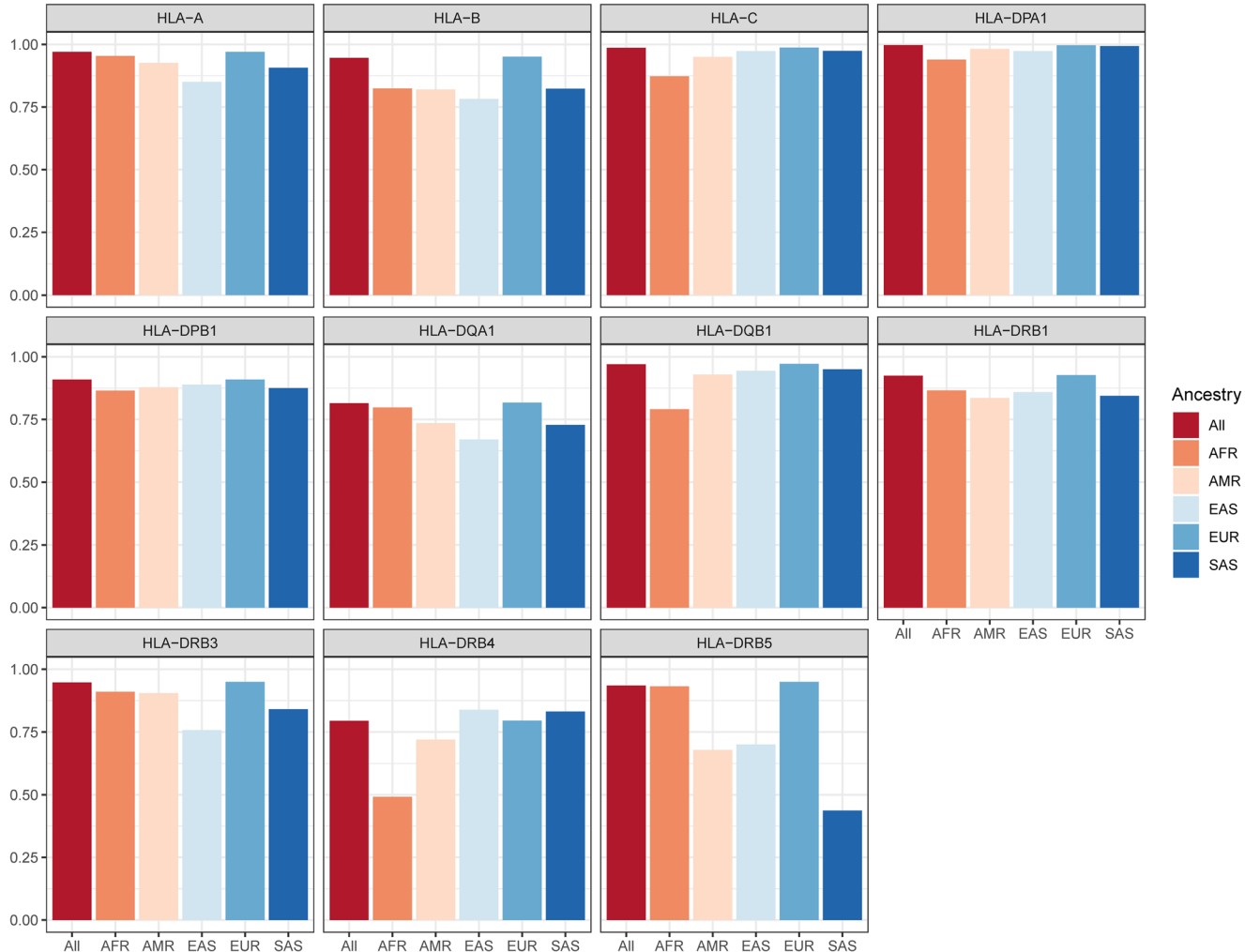

**Fig. 2 Adjusted concordance between sequenced and imputed HLA alleles.** Adjusted concordance (%) are shown stratified by HLA genes and genetic ancestry.

HLA allele imputation performance. This emphasizes the lack of diversity in the currently available imputation reference panels and limits the application of imputation-based approaches.

**HLA haplotypes LD**. To further describe our results and confirm their concordance with previous literature on the HLA, we characterized the LD pattern in the UKB cohort's HLA locus. Since HLA genes are multiallelic, the usual biallelic LD metrics give an incomplete portrait of the LD between pairs of HLA genes, as these would only provide pairwise HLA allele LD. Hence, we used an extension of biallelic LD that averages conditional LD measurements over the distribution of HLA alleles between pairs of HLA genes (asymmetric LD[23]). It has been used successfully in previous HLA studies and reduces to the standard $R^2$ measure of LD in cases where both variants (here HLA genes) are biallelic[23]. For this analysis, we used the 2-field HLA allele resolution to mitigate the effect of rare alleles, which makes the asymmetric LD calculation unstable (see Methods). While there were some variations between genetic ancestries in HLA haplotype LD patterns (Supplementary Fig. 5), most haplotypes in high LD were located in physical proximity to each other, with the following groups being closely associated across all genetic ancestries: 1) *HLA-B* and *HLA-C*, 2) the Class I genes (excluding *HLA-B* and *HLA-C*), 3) the *HLA-DR*, and *HLA-DQ* genes, 4) *HLA-DPA1* and *HLA-DPB1*, and lastly, 5) *HLA-DMA* and *HLA-*

*DMB*. Full asymmetric LD results per ancestry are provided in Supplementary Data 12–17.

**Allele frequency comparison to reference panel**. As the last quality check, we compared the allele frequencies obtained from WES HLA allele calling with those reported in the Allele Frequency Net Database (AFND)[24]. The AFND aggregates allele frequencies from multiple large cohorts, which we matched to the UKB biobank cohort based on their reported ancestries and country of origin (see Methods). However, given the sparsity of high-quality data on non-classical HLA genes in the AFND, we restricted this comparison to classical HLA genes (*HLA-A, HLA-B, HLA-C, HLA-DPA1, HLA-DPA2, HLA-DQA1, HLA-DQB1, HLA-DRB1*). Correlation between allele frequencies in the UKB and allele frequencies in the selected reference cohorts was high, suggesting that WES HLA calling performed well ($R^2$: 0.83; F-test $p$-value: $2.2 \times 10^{-16}$; intercept: 0.001, 95% CI: 0.0008–0.002; slope: 0.99, 95% CI: 0.98–1.01; Supplementary Fig. 6).

**HLA association studies in 11 auto-immune phenotypes**. To demonstrate the power of WES-based HLA analysis, we performed allele association studies for 11 phenotypes known to be associated with HLA genes: ankylosing spondylitis[25], asthma[26], autoimmune thyroid disorders[27], coeliac disease[28], Crohn's disease[29], type I diabetes mellitus[30], multiple sclerosis and other

demyelinating diseases (MS-Demyelinating)[31], polymyalgia rheumatica or giant cell arteritis (PMR-GCA)[32], psoriasis[33], rheumatoid arthritis[34], and ulcerative colitis[35] (Supplementary Data 18). The choice of these phenotypes was determined by the fact that they had previously established HLA associations (which would provide us with positive control associations), their large sample size in the UK Biobank, and their previous UK Biobank GWAS associations in the HLA regions (which ensure appropriate sample size for our analyses). Analyses were performed with Regenie[36] (see Methods) for each ancestry separately if there were 50 or more cases using age, sex, and the first 10 genetic principal components (PCs) as covariates (Supplementary Data 19). Ancestry-specific results were the meta-analyzed using fixed-effect meta-analysis with METAL[37] (Supplementary Data 20 for full summary statistics). We used a genome-wide significant threshold divided by the number of phenotypes as our statistical significance threshold (i.e., $p < 5 \times 10^{-8}/11$).

Our meta-analyses yielded 360 HLA allele associations at 3-field resolution (Table 1 and Supplementary Fig. 7), of which 118 were from HLA Class I genes. A pertinent positive control association is $HLA\text{-}B^*27:05$, an allele used for diagnosis and prognostication in clinical medicine[38], and which was highly associated with ankylosing spondylitis (OR: 6.55, 95% CI: 5.97–7.18, $p$: $1.97 \times 10^{-305}$, effect allele frequency [EAF]: 3.9%). Crohn's disease was the phenotype with the least associations (1), but the other phenotypes averaged 35.9 associations at the 3-field resolution, with the highest number of associations in autoimmune thyroid disorders ($n = 69$), coeliac disease ($n = 63$), and psoriasis ($n = 62$). To test how many of these associations were novel, we used the HLA-SPREAD PubMed abstract natural language processing database[39]. An association was considered previously reported if we could find it in the database. As an additional novelty check, we also repeated HLA association analyses using the imputed HLA alleles since these were already available and used in published association studies (even if they may not have been reported at all). Since most of the HLA literature restricts their analysis to 2-field precision, we used our 2-field association results and checked if these had been previously reported for their given phenotypes. The 2-field resolution analyses yielded 341 allele associations, of which 129 were likely novel. Of the rest, 44 were reported in the HLA-SPREAD database, while 168 could also be found using HLA allele association studies using the UKB imputed alleles (Fig. 3 and Supplementary Data 20).

Importantly, 103 of the 360 associations with 3-field resolution alleles were found in genes for which HLA imputation results were unavailable, suggesting that WES-based HLA allele calling could help discover many more HLA associations than previously possible. Moreover, many of these exhibited strong associations, both in terms of small $p$-values and large effect sizes, even in genes which were not previously known for a high degree of polymorphism. For example, $HLA\text{-}G^*01:06:01$ showed a strong association with psoriasis (OR: 1.80, 95% CI: 1.70–1.90, $p$: $3.57 \times 10^{-100}$, EAF: 6.2%). We also found multiple associations in rare alleles (MAF < 1%), including the novel $HLA\text{-}B^*57:31:02$ in psoriasis (OR = 4.61, 95% CI: 3.22–6.59, $p = 7.1 \times 10^{-17}$, EAF = 0.06%) and $HLA\text{-}C^*02:178$ in ankylosing spondylitis (OR = 5.33, 95% CI: 3.37–8.41, $p = 7.75 \times 10^{-13}$, EAF = 0.2%). While the $HLA\text{-}B^*57$ and $HLA\text{-}C^*02$ allele groups as a whole are already known to be associated with these diseases[40], this is, to our knowledge, the first time these specific alleles are reported. Given their high effect sizes, we believe that using WES for HLA allele calling in rare variants can allow us to better characterize the specific HLA variants and amino acid residues responsible for in risk of some diseases (here psoriasis and ankylosing spondylitis).

Many of these associations are unlikely to be observed due to HLA haplotype LD. In fact, of the 64,620 pairs of statistically significant 3-field resolution alleles (360*359/2), only 123 show a (biallelic) LD $R^2$ of 0.2 or more (Supplementary Data 21). More specifically, the $HLA\text{-}G^*01:06:01$ allele was only in mild LD with two other alleles associated with psoriasis: $HLA\text{-}A^*01:01:01$ ($R^2 = 0.21$) and $HLA\text{-}H^*02:01:01$ ($R^2 = 0.26$), which were both in high LD together ($R^2 = 0.71$) and less significantly associated with psoriasis than was $HLA\text{-}G^*01:06:01$ ($p = 1.77 \times 10^{-49}$ for $HLA\text{-}A^*01:01:01$, and $p = 1.53 \times 10^{-49}$ for $HLA\text{-}H^*02:01:01$). Hence, with respect to the other alleles included in this analysis, $HLA\text{-}G^*01:06:01$ was independently associated with psoriasis. While $HLA\text{-}G$ has been linked with other skin diseases, to our knowledge, associations between skin or soft tissue phenotypes and the $HLA\text{-}G^*01:06$ haplotype have only been reported in squamous intraepithelial cancer and cervical cancer[41].

Likewise, conditional analyses showed that 116 of the 129 novel allele associations were still significant after conditioning the most significant alleles of each phenotype (false discovery rate < 5%). Further, we found an additional 366 significant novel variant associations (false discovery rate < 5%) when conditioning on the most significant alleles for each phenotype, again suggesting that WES has increased power compared to imputation and supporting the validity of the likely novel allele associations found above. See Supplementary Data 22 for full conditional analyses results.

Lastly, among phenotypes which were analyzed in more than one ancestry and could therefore be meta-analyzed, there were no strong signal of heterogeneity in HLA associations. Indeed, qq-Plots of heterogeneity $p$-values show that there were no heterogeneous effects using our genome-wide threshold ($p < 5 \times 10^{-8}/11$), and only one association with p-value lower than the Bonferroni threshold ($p < 0.05/13,576$) at $DOB^*01:02:01$ for type 1 diabetes mellitus (see Supplementary Data 20 and Supplementary Fig. 8).

In summary, these findings suggest that WES-based HLA calling can identify many novel HLA-disease associations, possibly with large effects, compared to those identified through imputation-based approaches.

**Replication analyses in the Estonian biobank.** We performed the same analyses with 2-field imputed HLA alleles from the Estonian Biobank (Supplementary Data 23). Of the 341 allele associations found above, 196 were imputed and available for analyses. Of those 196 alleles, 123 were of the same effect direction and had a p-value less than the Bonferroni correction ($p < 0.05/196$) and another 25 had a $p$-value less than 0.05 (but above the Bonferroni correction). There was one additional allele with a $p$-value less than the Bonferroni correction and with an opposite effect direction. As suspected, of the potentially novel associations found above, only 8 were found in the list of imputed alleles, only 2 of which had a p-value less than 0.05 in the Estonian Biobank results. This again supports that WES-based HLA calling provides reliable and more accurate results than imputation-based methods.

**Effect of synonymous variants.** Given the increased allele resolution provided by WES-based HLA calling, we examined the effect of the additional HLA field on phenotype associations (i.e., from 2-field to 3-field resolution, wherein 3-field resolution would capture synonymous variants). We would expect similar and same-direction effect sizes for all synonymous variants in the HLA allele if they did not impact the phenotype. For example, if $HLA\text{-}A^*01:01$ was associated with a given phenotype, we would

**Table 1 3-field HLA allele association studies meta-analyses results.**

| Marker | Beta (log-Odds Ratio) | Standard Error | P-value | Direction | AF |
|---|---|---|---|---|---|
| Ankylosing Spondylitis (EUR) | | | | | |
| HLA-B*27:05:02 | 1.88 | 0.05 | $5.86 \times 10^{-305}$ | + | 0.04 |
| HLA-C*02:02:02 | 1.31 | 0.06 | $2.76 \times 10^{-94}$ | + | 0.03 |
| HLA-C*01:02:01 | 1.33 | 0.07 | $1.65 \times 10^{-92}$ | + | 0.03 |
| HLA-B*07:02:01 | −0.48 | 0.06 | $1.50 \times 10^{-17}$ | − | 0.14 |
| HLA-DRB1*01:03:01 | 0.81 | 0.10 | $4.81 \times 10^{-15}$ | + | 0.02 |
| Asthma (AFR, AMR, EAS, EUR, SAS) | | | | | |
| HLA-DRB1*04:01:01 | 0.20 | 0.01 | $1.69 \times 10^{-96}$ | +−−++ | 0.11 |
| HLA-DRB8*01:01 | 0.09 | 0.005 | $7.79 \times 10^{-74}$ | +++++ | 0.57 |
| HLA-DRB7*01:01:01 | 0.08 | 0.005 | $5.75 \times 10^{-64}$ | +++++ | 0.48 |
| HLA-DRA*01:01:01 | 0.08 | 0.007 | $1.35 \times 10^{-30}$ | +++++ | 0.57 |
| HLA-DRB1*13:01:01 | −0.17 | 0.01 | $1.38 \times 10^{-30}$ | −−+−− | 0.05 |
| Auto-immune thyroid disorders (AFR, AMR, EAS, EUR, SAS) | | | | | |
| HLA-DQA1*01:02:01 | −0.22 | 0.01 | $4.04 \times 10^{-90}$ | +−+−− | 0.17 |
| HLA-DRB1*15:01:01 | −0.20 | 0.01 | $7.11 \times 10^{-60}$ | +++−+ | 0.14 |
| HLA-DRA*01:02:03 | −0.20 | 0.01 | $9.34 \times 10^{-59}$ | ++−−− | 0.14 |
| HLA-DRA*01:01:01 | 0.14 | 0.009 | $1.32 \times 10^{-53}$ | +++++ | 0.57 |
| HLA-DRB1*04:01:01 | 0.19 | 0.01 | $2.73 \times 10^{-49}$ | ++++− | 0.11 |
| Coeliac (EUR) | | | | | |
| HLA-DQB1*02:01:08 | 0.96 | 0.04 | $6.42 \times 10^{-110}$ | + | 0.05 |
| HLA-DRB1*03:147 | 0.70 | 0.03 | $2.40 \times 10^{-91}$ | + | 0.10 |
| HLA-DRB7*01:01:02 | 0.70 | 0.04 | $3.13 \times 10^{-77}$ | + | 0.09 |
| HLA-H*02:01:01 | −0.72 | 0.04 | $9.88 \times 10^{-73}$ | − | 0.21 |
| HLA-DQB1*02:80 | 1.05 | 0.07 | $1.05 \times 10^{-55}$ | + | 0.02 |
| Crohn's (EUR, SAS) | | | | | |
| HLA-DRB1*01:03:01 | 1.03 | 0.08 | $9.97 \times 10^{-41}$ | +− | 0.016 |
| Diabetes mellitus (type 1) (AFR, EUR, SAS) | | | | | |
| HLA-DRB1*04:01:01 | 0.72 | 0.03 | $1.53 \times 10^{-119}$ | ++− | 0.11 |
| HLA-DQB1*03:02:01 | 0.77 | 0.03 | $1.69 \times 10^{-117}$ | +++ | 0.08 |
| HLA-DRB7*01:01:01 | 0.30 | 0.02 | $2.27 \times 10^{-72}$ | +++ | 0.48 |
| HLA-DRB8*01:01 | 0.27 | 0.02 | $3.62 \times 10^{-56}$ | +++ | 0.57 |
| HLA-DRB5*01:01:01 | −0.60 | 0.04 | $4.20 \times 10^{-54}$ | −−+ | 0.13 |
| MS-Demyelinating (EUR) | | | | | |
| HLA-DRB1*15:01:01 | 0.78 | 0.04 | $1.05 \times 10^{-108}$ | + | 0.144 |
| HLA-DRA*01:02:03 | 0.77 | 0.04 | $2.32 \times 10^{-105}$ | + | 0.146 |
| HLA-DRB5*01:01:01 | 0.87 | 0.04 | $3.05 \times 10^{-98}$ | + | 0.134 |
| HLA-DQA1*01:02:01 | 0.64 | 0.03 | $4.89 \times 10^{-82}$ | + | 0.167 |
| HLA-DRB5*01:100 | 0.84 | 0.05 | $7.39 \times 10^{-62}$ | + | 0.063 |
| PMR-GCA (EUR, SAS) | | | | | |
| HLA-DRB1*04:04:01 | 0.88 | 0.0409237 | $8.09 \times 10^{-102}$ | ++ | 0.04 |
| HLA-DRB7*01:01:01 | 0.27 | 0.0164555 | $2.56 \times 10^{-59}$ | ++ | 0.49 |
| HLA-DQB1*03:02:01 | 0.54 | 0.0338829 | $2.31 \times 10^{-57}$ | ++ | 0.08 |
| HLA-DRB8*01:01 | 0.24 | 0.0170413 | $3.47 \times 10^{-44}$ | ++ | 0.57 |
| HLA-DRB1*04:01:01 | 0.40 | 0.0312209 | $8.22 \times 10^{-37}$ | ++ | 0.11 |
| Psoriasis (EUR, SAS) | | | | | |
| HLA-C*06:02:01 | 0.99 | 0.02 | $1.50 \times 10^{-305}$ | ++ | 0.01 |
| HLA-B*57:01:01 | 1.10 | 0.03 | $1.91 \times 10^{-305}$ | ++ | 0.04 |
| HLA-DRA*01:01:02 | 0.89 | 0.03 | $7.32 \times 10^{-206}$ | ++ | 0.04 |
| HLA-DOB*01:05 | 0.78 | 0.03 | $7.73 \times 10^{-127}$ | ++ | 0.03 |
| HLA-G*01:06:01 | 0.59 | 0.03 | $3.57 \times 10^{-100}$ | ++ | 0.06 |
| Rheumatoid arthritis (AFR, AMR, EUR, SAS) | | | | | |
| HLA-DRB1*04:01:01 | 0.43 | 0.02 | $1.81 \times 10^{-96}$ | ++++ | 0.11 |
| HLA-DRA*01:01:01 | 0.27 | 0.02 | $4.51 \times 10^{-63}$ | +−++ | 0.57 |
| HLA-DRB7*01:01:01 | 0.17 | 0.01 | $8.58 \times 10^{-57}$ | −−++ | 0.48 |
| HLA-DRB4*01:03:01 | 0.22 | 0.02 | $2.97 \times 10^{-39}$ | −−++ | 0.16 |
| HLA-DQB1*03:02:01 | 0.31 | 0.02 | $1.27 \times 10^{-38}$ | ++++ | 0.08 |
| Ulcerative colitis (AFR, EUR, SAS) | | | | | |
| HLA-DRB1*01:03:01 | 0.97 | 0.06 | $7.42 \times 10^{-66}$ | +++ | 0.02 |
| HLA-DRB8*01:01 | −0.14 | 0.02 | $4.09 \times 10^{-20}$ | −−− | 0.57 |
| HLA-DRB7*01:01:01 | −0.13 | 0.02 | $4.57 \times 10^{-17}$ | −−− | 0.48 |
| HLA-DRB1*04:04:01 | −0.44 | 0.06 | $3.50 \times 10^{-15}$ | −−+ | 0.04 |
| HLA-DQB1*03:02:01 | −0.27 | 0.04 | $1.10 \times 10^{-11}$ | −−− | 0.08 |

Only five most significant results for each phenotype are shown (if there were more than 5 with *p*-value < $5 \times 10^{-8}$/11). Beta on logistic scale. Direction refers to effect direction for each ancestry in the meta-analyses (+ for positive, − for negative). The genetic ancestries used in each analysis are listed in parentheses after the phenotype name. Refer to Supplementary Data 20 for full summary statistics, including for 2-field accuracy, and by ancestry. *AF* Allele frequency.

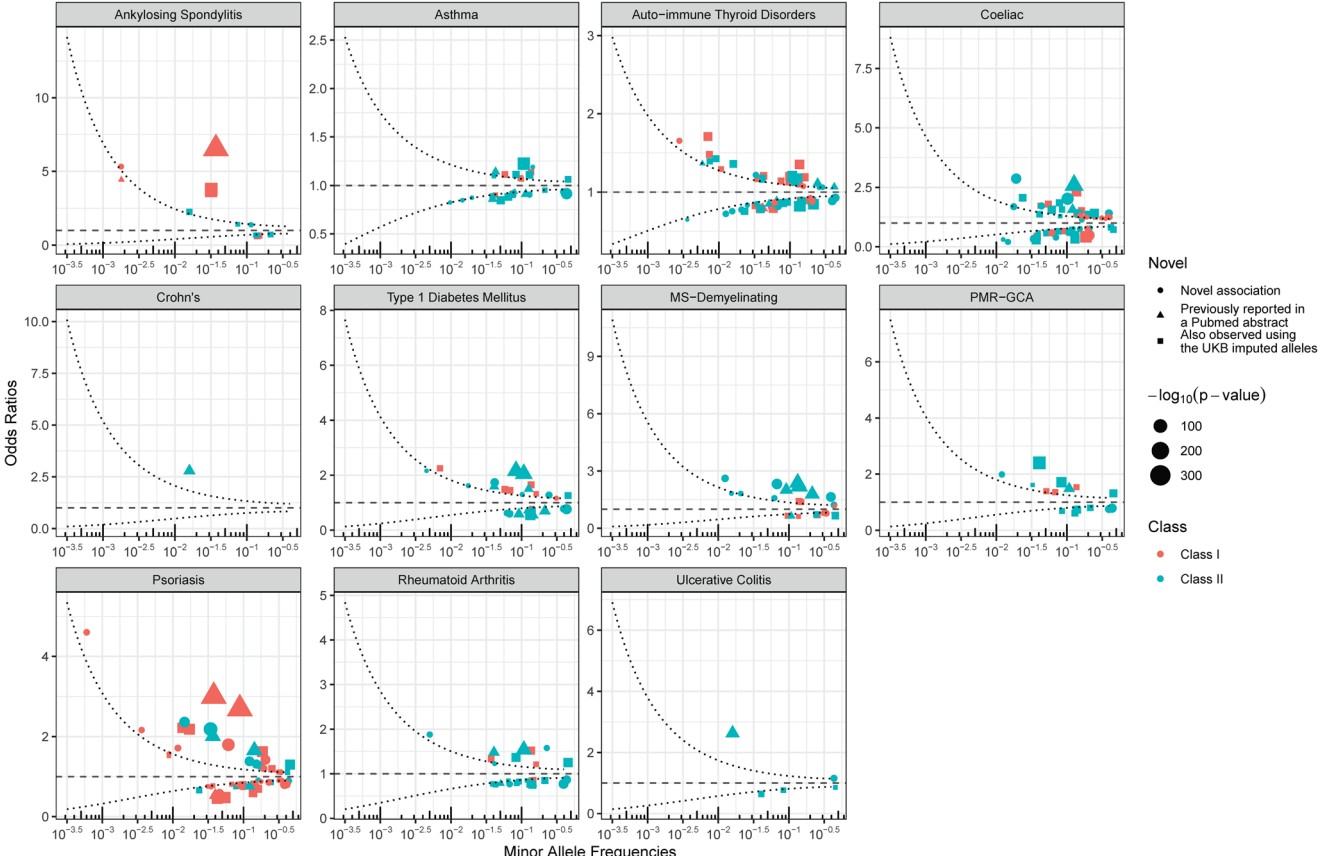

**Fig. 3 HLA association studies at 2-field resolution.** Odds ratios are shown. The curved dashed lines represent a power of 80% to detect an association at a $p$-value of $(5 \times 10^{-8})/11$ or less. Circles show likely novel allele associations. Pertinent positive controls include *HLA-B*27:05* for ankylosing spondylitis (OR: 6.55, 95% CI: 5.97–7.18, $p$: $1.97 \times 10^{-305}$, EAF: 3.9%). Also note the rare and novel allele associations *HLA-B*57:31* for psoriasis (OR: 4.61, 95% CI: 3.22–6.59, $p = 7.1 \times 10^{-17}$, EAF = 0.06%), and *HLA-C*02:178* for ankylosing spondylitis (OR = 5.33, 95% CI: 3.37–8.41, $p = 7.75 \times 10^{-13}$, EAF = 0.2%). For 3-field resolution results, refer to Supplementary Fig. 7.

expect *HLA-A*01:01:01* and *HLA-A*01:01:02* to show similar effects on the phenotype.

Specifically, we examined how synonymous variants in HLA alleles were associated with the 11 auto-immune phenotypes (Fig. 4). First, for any given phenotype, we found all 3-field resolution alleles that showed statistically significant association with a phenotype and compared their effect sizes with all other alleles of the same 2-field haplotype, hence directly isolating the effect of synonymous variants. We used Welch's t-test for unequal variances to compare effect size heterogeneity. We found 87 pairs of 3-field resolution alleles sharing the same 2-field haplotype, 54 of which showed a significantly different effect size ($p < 0.05/307$, see Methods). Of those, 11 were of opposite directions. For example, *HLA-DQB1*02:01:01* was associated with a 1.09-fold increase in odds of asthma (95% CI: 1.06–1.12, $p = 1.25 \times 10^{-10}$), *HLA-DQB1*02:01:08* was associated with a 0.90-fold decrease in risk (95% CI: 0.87–0.93, $p = 5.90 \times 10^{-11}$). There were an additional 42 pairs where one 3-field resolution allele was associated with the phenotype, but the remaining were not, and the heterogeneity was considered significant. For example, the *HLA-DQA1*01:02:01* allele was associated with a 0.94-fold decrease in odds of asthma (95% CI: 0.92–0.95, $p = 3.33 \times 10^{-16}$), but *HLA-DQA1*01:02:02* was not shown to be significantly associated (OR: 1.93, 95% CI: 0.99–1.06, $p = 0.01$). Lastly, 1 pair showed a difference in risk of disease in the same direction but with a different effect size: *HLA-DQB1*02:01:01* was associated with a 1.29-fold increase in the odds of coeliac disease (95% CI: 1.19–1.40, $p = 5.27 \times 10^{-10}$), but

*HLA-DQB1*02:01:08* was associated with an even higher risk with an odds ratio of 2.60 (95% CI: 2.39–2.87, $p = 6.42 \times 10^{-110}$). Of note, both alleles had relatively similar frequencies (6.5% and 4.7%, respectively).

However, many of the comparisons of these allele pairs may have been underpowered due to low allele frequencies. Therefore, when there were more than two 3-field resolution alleles with no association evidence for a given 2-field haplotype, we collapsed them all into a single 3-field resolution allele. This is conceptually the same process as a burden test (also known as a collapsing test) used for rare variant analyses[42]. In doing so, we aggregated enough alleles without association evidence to perform 220 additional pairwise comparisons, of which 12 suggested that there was a difference between the lead 3-field resolution allele and the corresponding collapsed HLA alleles. This included two instances where the effect was in the opposite direction, suggesting that at least one of the constituent 3-field resolution alleles in the collapsed allele was in the opposite direction as the lead 3-field resolution allele. For example, *HLA-G*01:01:02* was associated with a 0.70-fold decrease in odds of coeliac disease (95% CI: 0.65–0.75, $p$: $1.72 \times 10^{-20}$), while the burden test of its dummy 3-field allele showed an opposite direction (OR: 1.29, 95% CI: 1.21–1.39, $p$: $7.98 \times 10^{-13}$). This suggests that at least one synonymous variant in *HLA-G*01:01* obviates the association of *HLA-G*01:01:02* with coeliac disease.

In conclusion, the observed heterogeneity in the effects of synonymous variants at HLA alleles suggests that this type of variants is likely to contribute to the risk of human immune-

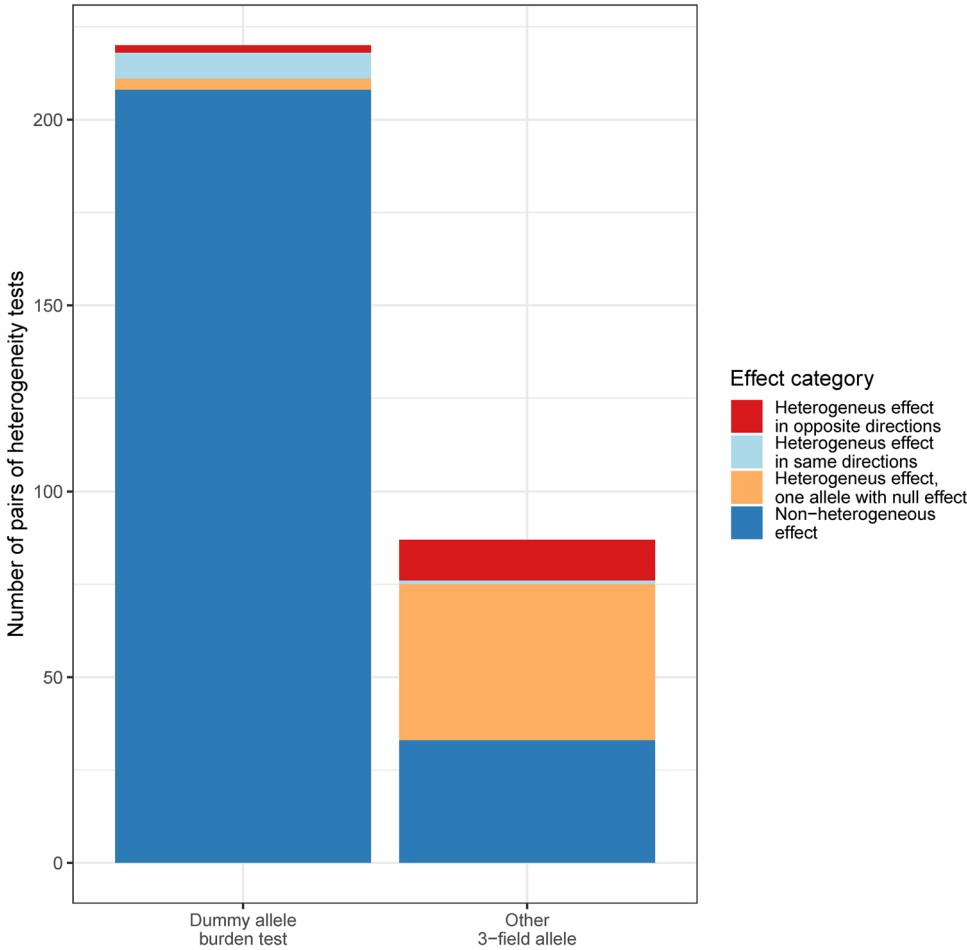

**Fig. 4 Summary of synonymous variant heterogeneous effect on HLA allele associations.** Result of pairwise effect heterogeneity tests for a 3-field allele to test for the effect of synonymous variants on traits. We compared the effect size of all 3-field alleles associated with any of the 11 phenotypes, with any other available 3-field allele falling in the same 2-field HLA haplotype (e.g., *HLA-A*01:01:01 and *HLA-A*01:01:02). Other 3-field allele: direct effect heterogeneity tests between pairs of 3-field HLA alleles. Dummy allele burden test: effect heterogeneity tests between pairs of 3-field HLA allele and all other 3-field alleles at that 2-field haplotypes combined in a dummy allele. Colours represent if the heterogeneous effect is due to each pair having an opposite effect direction (red), the same effect direction (beige), or one of the allele pairs having a non-significant association with the given phenotype (light blue).

mediated diseases. To our best knowledge, most previous HLA fine-mapping studies were limited to 2-field resolution alleles and did not capture synonymous variants' effects. Specifically, we are not aware of studies of comparable size that studied the effect of synonymous HLA variants in a systematic way. See Supplementary Data 24 for the complete results of these analyses.

**Imputed vs sequenced HLA alleles canonical correlation analysis (CCA) and PRSs.** While our prior results supported the hypothesis that WES-based HLA allele calling would be more accurate than imputation, imputation may still provide enough information to still be adequate for PRSs, even if it does not assign alleles accurately to participants. To test this hypothesis, we first used CCA on the matrices of imputed and WES-based HLA alleles (using a 0, 1, and 2 encoding for absent, heterozygous, and homozygous for the allele). CCA performs linear transformations of both matrices to find the best linear approximation of one against the other. In other words, it assumes the existence of a set of variables that both HLA imputation and WES-based HLA calling approximate, allowing for the calculation of the amount of variation in WES-based HLA alleles that can be explained by imputed HLA alleles (also referred to as total canonical redundancies). If the variance explained by the imputed alleles is high,

we would not expect a great increase in PRS predictive ability from using WES-based HLA alleles. Indeed, using CCA (Supplementary Data 25 and Supplementary Fig. 9) for WES-based HLA alleles with AF > 10%, we found that imputed HLA alleles can account for 85.1% of the variation in WES-based alleles. This increased to 88% with AF > 20%. This decreases when we lower the AF threshold (e.g., decreases to 77.5% for AF > 5%) or add the non-imputed genes because these are not captured well (or at all) by imputation. Additionally, as expected, the percentage of variation in WES-based alleles explained by imputed alleles was driven by the EUR ancestry cohort. These values varied in other genetic ancestries, with the percentage of variance explained in the AFR ancestry cohort consistently lower than in other cohorts (except for the analysis with AF > 0.01% and only using the imputed genes). However, the considerable differences in sample size make further comparisons between genetic ancestries difficult or even impossible (e.g., the analyses could not be performed in the EAS ancestry cohort).

We then used the LDpred software to compute PRSs from the seven phenotypes for which complete GWAS summary statistics were available in the GWAS Catalog. All GWASs contained only participants of EUR genetic ancestry, except for rheumatoid arthritis. Two LDpred scores were obtained for each phenotype

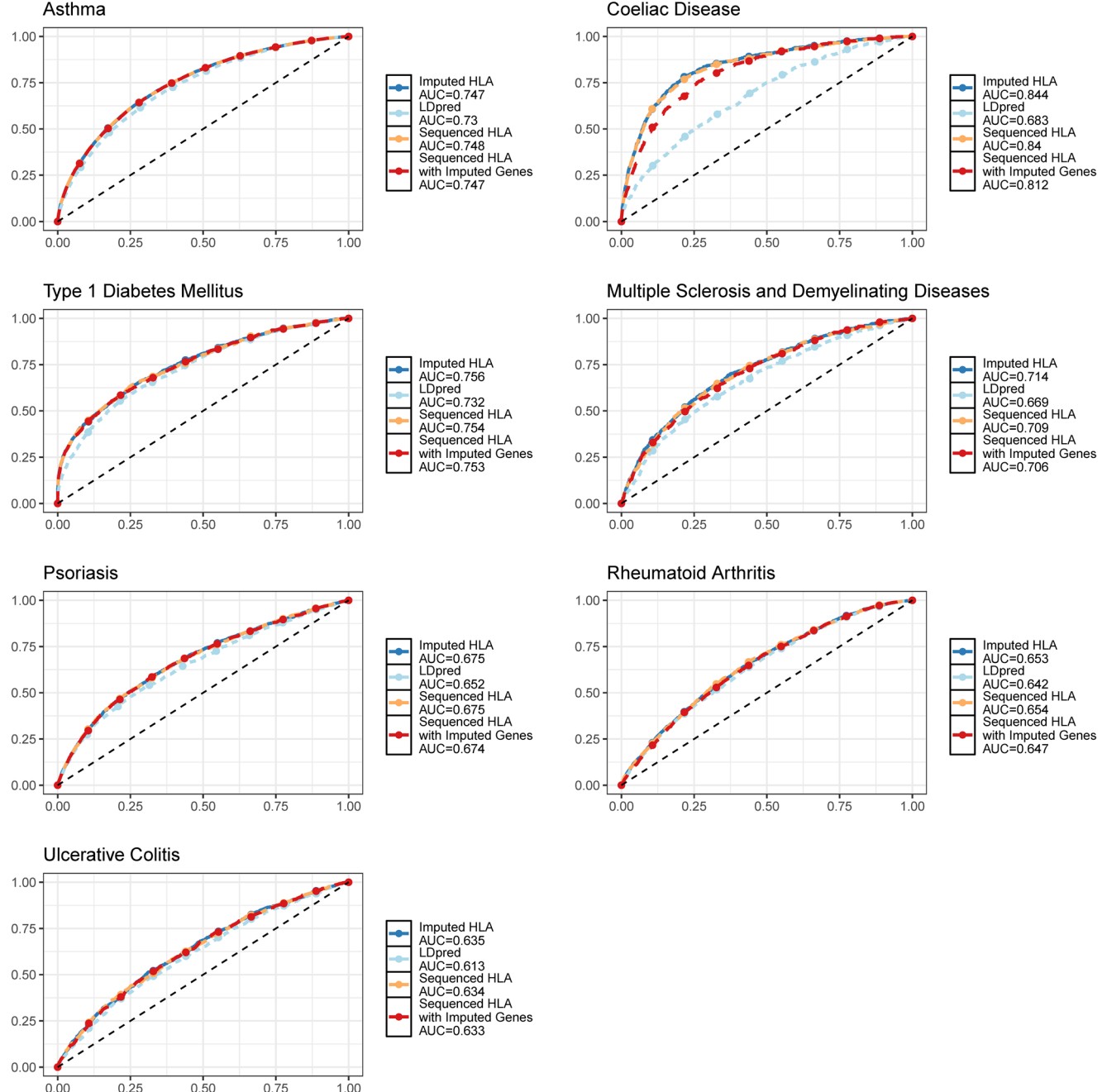

**Fig. 5 PRS performance for seven traits.** X-axis: false positive rate. Y-axis: true positive rate. Dashed line: 50% AUC representing random diagnostic performance (i.e., curves further away from the dashed lines represent better models). AUC Area under the curve. See main text for more information on the four PRS tested.

and each participant: 1) a score from the summary statistics across the whole genome and 2) another after removing the HLA region variants. The LDpred scores were then used in an XGBoost binary classifier for each phenotype, along with age, sex, and the first 10 PCs. For the classifier using the LDpred scores without the HLA region, we also added either the imputed HLA alleles or the WES-based HLA alleles in the XGBoost algorithm. Hence, we performed three distinct XGBoost PRSs for each phenotype: 1) LDpred scores alone, 2) LDscores without the HLA region but with the imputed HLA alleles, and 3) LDpred scores without the HLA region but with the WES-based HLA alleles. HLA alleles improved all PRSs to varying degrees with an average absolute increase in area under the receiver operating characteristic curve

(AUC) of 0.085 (for both alleles containing HLA alleles, see Fig. 5 and Supplementary Data 26). The largest increase was for coeliac disease: the AUC increased from 0.68 (95% CI: 0.66–0.70) to 0.84 (95% CI: 0.82–0.86) with WES-based HLA alleles (compared to not using HLA alleles). However, the difference between AUC of the PRSs using imputed and WES-based alleles was always small (average of 0.002), with all 95% confidence intervals for the difference in AUC containing zero (Supplementary Data 26). Hence, as expected, given our CCA results, HLA allele imputation explains enough of the variation in the UKB participants' HLA alleles to still be useful for PRS purposes. However, this may not hold for non-EUR genetic ancestry cohorts, given the limited diversity of available HLA imputation reference panels. Similar

results were seen with precision-recall curves (Supplementary Fig. 10).

In this study, HLA imputation was performed using the same imprecise algorithm in both the training and the test set. In real-world applications, one would use a PRS developed with the HLA alleles from one imputation algorithm in a separate population, probably using a different HLA typing method (e.g. another imputation software). To mimic this scenario, we used the XGBoost weights from our PRS developed with imputed HLA alleles, and we used WES-based HLA alleles for the testing set's input features. This did not lead to large differences in AUC. HLA imputation provided a small improvement except for coeliac disease (AUC decreased from 0.84 to 0.81, absolute difference: −0.03, 95% CI: −0.06 to −0.0004). We conclude that HLA allele imputation may be useful for PRS, but most phenotypes will not benefit beyond the inclusion of the tag variants in the HLA region of the genome.

**Amino acid association studies**. Using the 2-field allele calls, we performed amino acid residue fine-mapping at all protein-coding genes for all 11 phenotypes. The same analytic method was used as in the HLA allele association studies above. As expected, we found many more associations ($p < 5 \times 10^{-8}/11$) for amino acid residues than HLA alleles. We found 5,556 associations in our multi-ancestry meta-analyses, with 2134 for autoimmune thyroid disease, coeliac disease, or psoriasis (Supplementary Data 18). However, the correlation structure of those residues is significantly more complicated than for HLA alleles LD, making causal inference even more complex. For example, while it is clear that residue 57 of the HLA-DQB1 protein is the main determinant of type 1 diabetes mellitus at this gene (as reported before in refs. [43–45]), with an odds ratio of 1.72 (96% CI: 1.64–1.80, $p = 5.4 \times 10^{-116}$), it is not as clear which amino acid is the main driver at HLA-DQA1 (Fig. 6a). Nevertheless, these amino acid association studies can still provide important insights into the genetic underpinnings of the HLA, especially when considering potential interactions between amino acids. Specifically, the HLA-DQA1 and HLA-DQB1 proteins form a heterodimer and should be analyzed together, and when performing normal mode analysis of this heterodimer (to see which amino acid move together in space, see Fig. 6b) or when looking at the distance between amino acid in 3-dimensional space (Fig. 6c), it becomes clear that residues 53–57 of HLA-DQB1 are in close contact with amino acids 60–80 of HLA-DQA1. Notably, residues 60 to 80 are part of a segment of the HLA-DQA1 proteins (residues 45–80) with nearly identical p-values ($5.81 \times 10^{-58}$) and high LD. Indeed, these two segments are in close contact and are part of the ligand binding groove of the HLA-DQA1/HLA-DQB1 heterodimer (Fig. 6d). Hence, WES-based HLA allele calling and amino acid fine-mapping can provide additional biological evidence on the role of the HLA in human disease.

Lastly, in contrast to HLA alleles, we observed significant heterogeneity in amino acid associations for the auto-immune thyroid disorders, type I diabetes mellitus phenotypes, and asthma to a lesser extent (Supplementary Fig. 8). This was especially the case in class II HLA genes DOB, DRB1, DRB5, and DQB1 (Supplementary Data 20). This potentially represents amino acid residues which are neither causal, nor highly correlated with causal genetic variants, but more research would be needed to confirm this hypothesis.

## Discussion
Here we report an increased accuracy in WES-based HLA allele calling compared to imputation-based approaches. This gain in accuracy was greater for rare variants and non-EUR genetic

ancestry UKB participants. This improved accuracy allowed us to identify 360 allele associations at 3-field resolution for 11 auto-immune phenotypes. At 2-field resolution, we found 341 associations, of which 129 were likely novel. The increased resolution (from 2-field to 3-field) afforded by WES also allowed us to better characterize the association between synonymous variants in HLA alleles and human diseases. We found that for at least 25% of 2-field haplotypes exhibiting synonymous variants in the UKB, the resulting 3-field haplotypes showed statistically significant heterogeneous effect sizes. This observation and the fact that 2-field accuracy decreased the number of allele associations we found (from 360 at 3-field to 341 at 2-field) supports the hypothesis that the increase in accuracy also improved statistical power, as the collapse of multiple 3-field alleles into one 2-field allele would hide potential HLA allele disease associations. Given that previous HLA association studies usually do not consider the effect of synonymous variants, this represents an advance in our understanding of the HLA. Lastly, we showed that using HLA alleles from either imputation or sequencing may improve PRS accuracy, while WES-based HLA alleles may improve their external validity. More specifically, we showed that for some diseases, using a different method to assign alleles to participants in the training cohort than in the test cohort may decrease the accuracy of the PRS. This is likely even more important for non-EUR genetic ancestries, which are under-represented in HLA research. Hence, WES-based HLA allele calling provides additional insights into the complex role of the HLA in human diseases. These insights will likely be important for future translational research programs on the HLA and its application to therapeutic drug development. Importantly, these HLA allele calls for all UKB participants will be made available to the scientific community.

Our results highlight some limitations and areas for future research. First, the UKB WES program was not designed with the specific aim of HLA calling, and it uses a short read technology. It is known that HLA-specific assays with longer reads will perform better for this region, and an increased accuracy would be expected from such technology[46]. Additionally, as the UKB will release whole-genome sequencing data for its entire cohort (still only available for around 150,000 participants at the time of writing this manuscript), a comparison between WES and WGS will be needed, as the optimal trade-off between better non-coding region coverage and depth of sequencing in the HLA is unclear. Second, newer imputation algorithms have been developed since the UKB first released their HLA imputation results, and these may fare better with rare alleles and non-EUR ancestry individuals than our current comparator. Nonetheless, the current best-performing algorithm from the Michigan Imputation Server had a lower concordance rate than HLA-HD when compared with the 1000 G panel[18,47] (e.g., 100% concordance at HLA-DRB1 for HLA-HD, and between 90.9% to 96.9% for the Michigan Imputation Server, depending on ancestry, both at 2-field). This server also currently only provides imputation for 9 genes. Hence, using HLA sequencing when available appears preferable. Third, HLA-HD does not provide 4-field resolution, which would be necessary to study non-coding variants, including those that may be tagging synonymous variant, explaining the signal we found when comparing 2-field to 3-field. Given our findings on potential the non-negligible role of synonymous HLA variants in human disease, we expect that non-coding variants would also be important to study more thoroughly. The upcoming release of the full UKB participant whole-genome sequencing data should shed light on this issue. However, this will need the development of an HLA calling algorithm capable of handling 4-field resolution while also being scalable on cloud computing architecture.

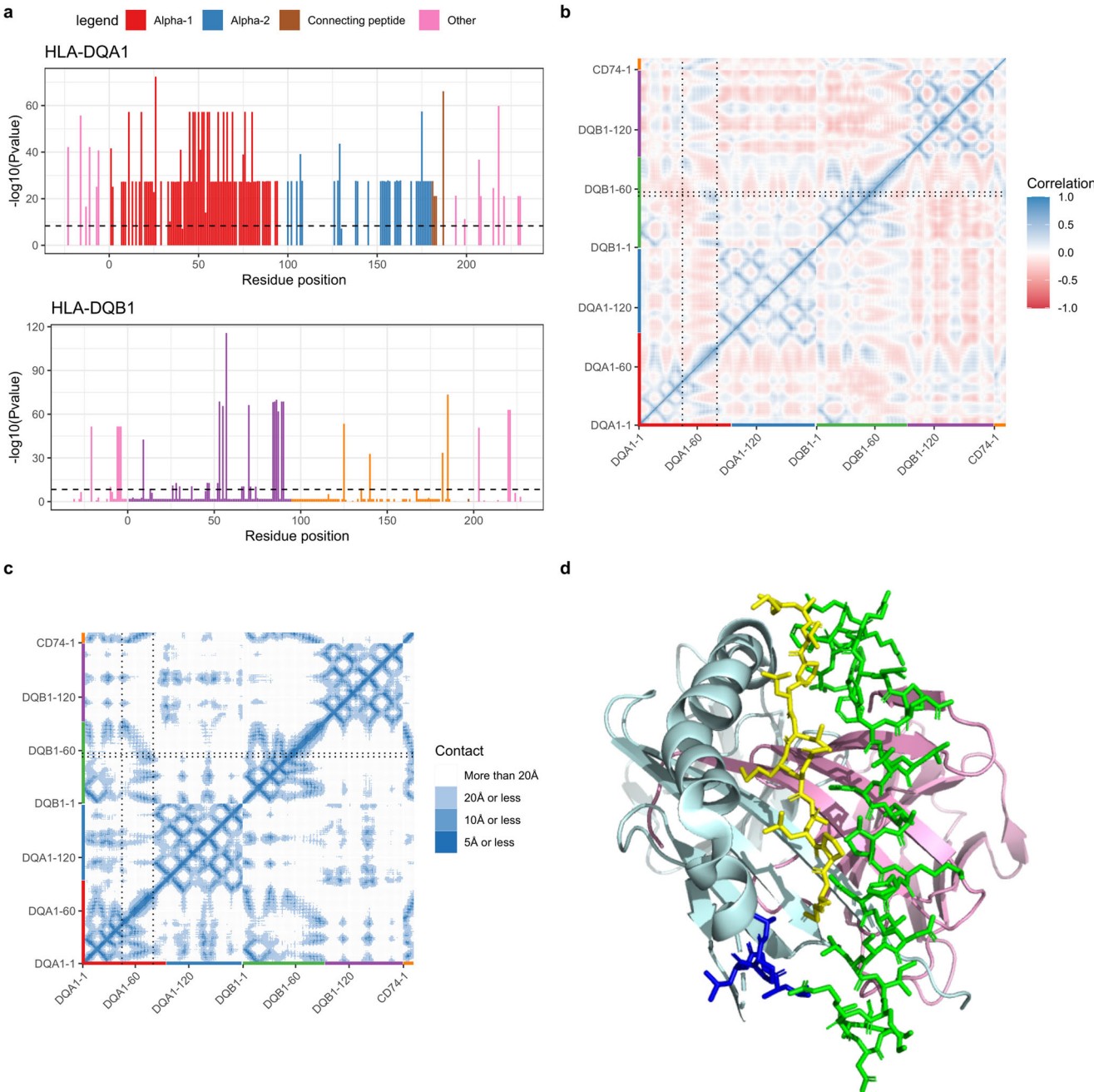

**Fig. 6 HLA-DQ heterodimer and its association with type I diabetes mellitus.** HLA-DQ heterodimer is encoded by *HLA-DQA1* and *HLA-DQB1*, and their interaction needs to be accounted for when studying amino acid residue associations, such as here for type I diabetes mellitus. **a** Summary of *HLA-DQA1* and *HLA-DQB1* type I diabetes mellitus association study. The X-axis represents residue positions, with colours corresponding to the chain they are a part of (e.g., alpha-1 chain of *HLA-DQA1*). The Y-axis represents the smallest *p*-value at each position (on the -log10 scale). Only residue positions with polymorphisms can be shown. As can be seen, the *HLA-DQA1* results likely suffers from the high correlation in amino acid residue inheritance as residues 45 to 80 all show a similar association with type I diabetes mellitus. This isn't affecting *HLA-DQB1*, where residue 57 (on the beta-1 chain) is the most significant. **b** Normal mode analysis of the HLA-DQ heterodimer shows that residues 53–57 of *HLA-DQB1* interact with some residues of the 45–80 HLA-DQA1 protein (the square formed by the 4 dotted lines), which is also observed in **c** The contact map. Colours on the contact map show distances. **d** A 3-dimensional view of the HLA-DQ heterodimer showing that residues 53–57 of HLA-DQB1 likely interact with the more distal residues of the HLA-DQA1 45–80 segment, which directly interacts with the HLA-DQ ligand binding. The blue segment represents residues 53–57 of HLA-DQB1. The green segment represents residues 45–80 of HLA-DQA1. The yellow segment is a ligand binding the dimer (here CD74). Pale green and pink cartoon protein representations show the rest of HLA-DQB1 and HLA-DQA1, respectively.

Fourth and most importantly, new and more accurate technology is only one piece of the puzzle to better understand the role of the HLA in diseases. There are still many unresolved questions relating to haplotypes LD and HLA protein interactions (since HLA proteins form complexes) that remain to be solved. In

the past, cross-ancestry comparative genetics has been used to better determine causal variants, and the study of the HLA would benefit from the sequencing of more non-EUR genetic ancestry individuals. Further, it was previously described that using HLA alleles instead of single nucleotide polymorphism increases the

yield of HLA genes in expression quantitative trait loci studies[48]. Hence, a large-scale association study of HLA allele determinants of HLA gene expression, splicing, and protein level using genome/exome sequencing and HLA calling algorithm would shed much-needed light on how HLA polymorphisms affect diseases. Finally, the best way to incorporate HLA alleles in PRS also deserves further research. For example, by modelling HLA allele interactions, previous literature showed improved PRS performance compared to the method we used in this paper[30]. It would be of interest to see how this translates using WES/WGS technology and HLA allele calling, especially in non-EUR ancestries.

In conclusion, using WES for HLA allele calling improves the accuracy of ascertainment and, therefore, statistical power to associate HLA alleles with diseases. This should help solve the problem of the HLA's highly polymorphic character and LD. Doing so is particularly important for genetic ancestries under-represented in research and for rare variants. This is important since, by increasing HLA typing accuracy, a better understanding of the genes responsible for diseases at HLA is possible, which could be translated into actionable therapies.

## Methods

**Statistics and reproducibility**. Many analyses were performed, the following method sections were written with the aim of facilitating its reproducibility. References to all data, software, code, sample size and statistical tests are described in details in the following sections.

**HLA allele sequencing**. We used WES data from 454,824 individuals in the UKB to call HLA alleles. First, CRAM WES alignment files were converted to fastq files using Picard tools[49] and the GRCh38 human reference genome[50]. Second, HLA-HD[18] (v1.4.0) was used to call all possible HLA alleles. For the allotted coverage in the WES data, this corresponded to the following 31 genes or pseudogenes (see resource 3803 of the UKB for target regions details): *HLA-A, HLA-B, HLA-C, HLA-E, HLA-F, HLA-G, HLA-H, HLA-J, HLA-K, HLA-L, HLA-V, HLA-Y, HLA-DMA, HLA-DMB, HLA-DOA, HLA-DOB, HLA-DPA1, HLA-DPA2, HLA-DPB1, HLA-DQA1, HLA-DQB1, HLA-DRA, HLA-DRB1, HLA-DRB2, HLA-DRB3, HLA-DRB4, HLA-DRB5, HLA-DRB6, HLA-DRB7, HLA-DRB8, HLA-DRB9*. HLA-HD uses Bowtie2[47] to align WES data to the reference genome. Only segments of 50 base pairs or longer were used, as the Bowtie2 aligner documentation recommended. We used the IPD-IMGT/HLA release 3.45.0. The entire pipeline was implemented on DNAnexus (Palo Alto, California, USA) using the Workflow Description Language. Two separate Docker[51] images were used in the workflow: 1) the broadinstitute/gatk[52] image to convert CRAM files to fastq, and 2) a Docker image containing HLA-HD and its dependencies for the HLA calling.

**HLA allele calls processing**. Final HLA allele calls were first transferred to our local computing cluster. While these files will be made available through the UKB return of data program, for the remainder of the analyses in this paper, we only used HLA calls with a total coverage of 20 reads at exon 2, except for *HLA-DRB2* and *HLA-DRB8*, where a total coverage of 20 reads at exon 3 was used since these two genes do not have a second exon.

We used an additional heuristic approach to assign alleles at the *HLA-DRB3, HLA-DRB4*, and *HLA-DRB5* genes. Like other HLA allele calling from sequencing technology algorithms, HLA-HD may provide calls for the *HLA-DRB3-4-5* genes if some reads aligned to one of these genes, even if a participant may not truly carry a copy of them (in which case this alignment would be incorrect). This is because while everyone has two copies of the

*HLA-DRB1* gene (maternal and paternal), each copy can sometimes (but not always) be inherited along with a copy of either *HLA-DRB3, HLA-DRB4*, or *HLA-DRB5*, for a total of 2 to 4 *HLA-DRB* genes: two *HLA-DRB1*, and a combination of zero to two of any of the other three genes. Hence, it is not sufficient to use the HLA-HD calls at these genes; one must also quantify the number of reads at these genes and compare them to a reference to decide which of these genes (if any, and how many) are carried by each individual. Here, a natural reference would be the quantity of *HLA-DRB1* measured in a participant since these genes are in high LD. Intuitively, a participant with a very low quantity of *HLA-DRB3-4-5* compared to *HLA-DRB1* should not carry any of the *HLA-DRB3-4-5* genes. On the other hand, if the quantity of *HLA-DRB3-4-5* is the same as that of *HLA-DRB1*, then the participant should have 2 of these genes (any combination of *HLA-DRB3, 4, or 5*). A similar logic applies to participants who carry only one copy of an *HLA-DRB3-4-5* gene, as they would be expected to have half as many reads at these genes than at *HLA-DRB1*. Hence, we compared the number of reads at exon 2 at the *HLA-DRB* genes to decide which one to assign and used this heuristics-based approach to assign alleles at *HLA-DRB3-4-5*. For every participant, if one of their *HLA-DRB3-4-5* allele calls had a total coverage of 60% of the *HLA-DRB1* coverage (and still more than 20), we used the two alleles for this gene as called by HLA-HD. For example, if a participant had a mean *HLA-DRB1* coverage of 100 and a mean *HLA-DRB4* coverage of 60, we assigned both *HLA-DRB4* alleles to this participant. Otherwise, if one or more of their alleles had a mean coverage of 30% or more of the *HLA-DRB1* coverage, we assigned them the first called allele by HLA-HD from the respective genes. For example, the same participant with an *HLA-DRB1* coverage of 100 having an *HLA-DRB4* coverage of 30 and an *HLA-DRB5* coverage of 30 would be assigned the first allele of each of these genes. If no alleles from *HLA-DRB3-4-5* fulfilled these conditions, the participant was assigned no alleles from those genes.

Using these allele calls, we also assigned amino acid polymorphisms at each position of the 19 protein-coding genes: *HLA-A, HLA-B, HLA-C, HLA-E, HLA-F, HLA-G, HLA-DMA, HLA-DMB, HLA-DOA, HLA-DOB, HLA-DPA1, HLA-DPB1, HLA-DQA1, HLA-DQB1, HLA-DRA, HLA-DRB1, HLA-DRB3, HLA-DRB4, HLA-DRB5*. We then converted these allele and amino acid calls into VCF[53] (with dummy positions) and PLINK[54] binary files for our analyses. For the 2-field HLA allele analyses, we trimmed the 3-field alleles to two fields by removing the 3rd field and the change in expression suffix (when present). All data processing was performed using R[55] (v4.1.0), BCFtools[56] (v1.11-1-g87d355e), and PLINK (v1.9).

**Genetic ancestry assignment and principal components**. We used the somatic chromosomes imputed genome-wide genotypes from the UKB to assign 1000 Genome continental genetic ancestry to every participant (AFR, AMR, EAS, EUR, and SAS). To do this, we first selected variants with minor allele frequency (MAF) > 10%, call rate > 95%, Hardy-Weinberg equilibrium $p$-value > $10^{-10}$, and which are part of the 1000 G reference panel. We trained a random forest classifier for genetic ancestries using the first 6 PCs of the 1000 G reference. We then projected the pruned UKB genotypes on the 1000 G reference PCs and assigned genetic ancestries using the majority call from our classifier.

To compute PCs to use as covariates in our association tests, we split participants by their genetic ancestry group and kept only variants with MAF > 1%, minimum allele count > 100, call rate > 95%, and Hardy-Weinberg equilibrium $p$-value > $10^{-10}$. These were then LD pruned with the $r^2 < 0.2$ threshold, and the resulting variants were used to obtain PCs. For the European

ancestry group, we used fast approximate PCs due to the large number of individuals, as implemented in PLINK[57] (v2.0).

All analyses and computations for this section were done using PLINK (v1.9 and v2.0), BCFtools, or R.

**Comparisons with UK Biobank HLA allele imputation**. HLA calls were compared to the available UKB imputed HLA alleles obtained with HLA:IMP*2[13] (data field 22182) for the following genes: *HLA-A, HLA-B, HLA-C, HLA-DQA1, HLA-DQB1, HLA-DPA1, HLA-DPB1, HLA-DRB1, HLA-DRB3, HLA-DRB4, HLA-DRB5*. Alleles ending in 99:01 were considered unimputed, and alleles with imputation dosage less than 0.8 were set to 0 as per UKB documentation. We also used a liberal threshold of 0.66 (the most liberal threshold ensuring a maximal number of alleles of 2 per genes), and a strict threshold of 0.9 for comparisons. The strategy and results of comparison between the imputed and sequenced alleles are presented in Supplementary Data 6–11. Individual-specific allele frequency was computed as the mean of the frequencies of the two corresponding alleles. For each gene, concordance was calculated by summing the number of matching alleles between imputed and called alleles across participants and dividing by twice the number of participants. Since this did not take changes in IMGT-HLA into accounts, we also calculated an adjusted concordance using only sequenced alleles that were also found in the imputation tool's reference database.

**HLA allele LD**. We computed multiallelic asymmetric LD[23] for each HLA gene. This was done for each ancestry separately as well as globally for the entire cohort. In each analysis, we assigned a dummy allele to unsequenced alleles and alleles with frequency < 1%. This analysis provides conditional LD for any pair of genes (e.g., LD of *HLA-A* conditional on *HLA-B*, and vice versa). We then took the average of the conditional LD pair to create a correlation heatmap clustered using the R hclust function with the "average" clustering method. The first 5 hclust clusters were highlighted by rectangles in the heatmaps. This section was done with R.

**Allele frequency comparison to reference panel**. To compare allele frequencies from WES HLA calling to reported reference frequencies, we used the AFND[24] to find cohorts with reported HLA allele frequencies and with genetic ancestries comparable to those in the UK Biobank. Specifically, for each 1000 Genome continental genetic ancestry (AFR, AMR, EAS, EUR, and SAS), we found a similar cohort in the AFND based on reported ancestry and country of origin. When multiple cohorts were available, we prioritized the one with the largest sample size. We only looked for cohorts with a sample size larger than 500 and reported as high quality ("gold population standard" option). Additionally, we only used cohorts if the sum of all reported allele frequencies for each gene was 1. Lastly, we used allele frequencies reported at an accuracy of 2-field since the largest high-quality cohorts did not report frequencies at 3-field. Note that this analysis was restricted to classical HLA genes, given the sparsity of high-quality data on other genes in the AFND. A complete list of AFND cohorts used as comparators can be found in Supplementary Data 27. Lastly, all allele frequency comparisons between the UKB WES HLA allele calls and the selected AFND reference cohort were made separately for each genetic ancestry.

**Phenotypes classifications**. We selected 11 phenotypes with known associations with HLA alleles to have multiple true positive and true negative results to check for in our association results. We used either ICD-10 codes from hospitalization electronic medical records data (data fields 41202 and 41204),

disease-specific data fields (e.g., data field 6152, option 8, for asthma), or self-reports (data field 20002), depending on the disease. The choices of data fields and ICD-10 codes (Supplementary Data 18) were based on previous studies[26,27,58] validating their use and reviewed by a board-certified physician (GBL). These data fields were aggregated directly on the DNAnexus web service.

**HLA allele and amino acid association studies**. Regenie[36] (v2.2.4) was used for all association tests (2-field HLA alleles, 3-field HLA alleles, and amino acids). Regenie works in two steps. In the first one, a risk score for the given phenotype is assigned to each chromosome for each participant by ridge regression. In this step, we used the ancestry-specific pruned whole-genome genotyped data, the same as for the ancestry-specific PCs described previously. In the second step, each chromosome score is used as a covariate in the association model to adjust for kinship structure and polygenic background. To avoid proximal contamination, this association model does not use the score of the chromosome where the variant is located (so-called LOCO scheme) since this score may already include the effect of the given variant. Hence, in Regenie's second step, we used PLINK binary files with assigned chromosome 6 and a dummy chromosomal position for each HLA allele and amino acid. This ensured that kinship was accounted for without adjusting for the effect of variants on chromosome 6 in the null model. Our association model also included age, sex, and the first 10 PCs as covariates. The approximate Firth regression method was used for all association tests to provide unbiased effect estimates even for rarer alleles and amino acids while accounting for case-control imbalance. For amino acids, we used alignment provided by the IMGT-HLA[59] to determine residue positions, and we excluded indel sequences from the analysis (i.e., those that correspond to either an insertion of additional amino acid residues to the protein or to a deletion of residues from the same protein). Other specific Regenie parameters included a minimal case count of 50, a genotype size blocks of 1000 in step 1 and 400 in step 2 (based on Regenie's UKB documentation), a Firth regression p-value threshold of 0.1 with back-corrected standard error (--firth-se), a minimum allele count of 1, and the --htp option. All analyses were done separately per ancestry, then meta-analyzed using fixed effect inverse variance weights in METAL[37]. For the statistical significance threshold, we mimicked the common situation where a researcher performs the HLA association study following the result of a GWAS with a locus at the HLA. Hence, we used the usual $5 \times 10^{-8}$ genome-wide significance threshold[60] divided by 11 (the number of phenotypes used). Given the use of rare alleles which may lead to unstable p-values and an increase in type 1 error, we also used negative controls phenotypes (osteoporosis, atrial fibrillation, non-insulin dependent diabetes mellitus) to ensure that this threshold and the Firth regression was enough to control the type 1 error rate. These phenotypes are expected to have no strong HLA association, except for non-insulin dependent diabetes mellitus which is often contaminated by Type 1 diabetes mellitus cases in large registries. As expected, none of these phenotypes showed significant HLA associations with our HLA alleles at $p < 5 \times 10^{-8}/11$ (see Supplementary Fig. 11 and Supplementary Data 28).

**Determining if a sequenced HLA allele association was novel**. We used two sources to determine if an allele association was novel. We first used the HLA-SPREAD database[39], which used a natural language processing algorithm to scan 28 million PubMed abstracts for HLA allele associations. If an allele association was reported in the database, it was not deemed novel. For the

remaining potentially novel associations, we performed the same HLA allele association analysis as above but used the imputed HLA alleles instead of the WES-based ones. If an allele was genome-wide significant in both ($p < 5 \times 10^{-8}/11$), the WES-based HLA allele association was considered potentially not novel, as it could have been reported in a previous study using the UKB, though we recognize that we may still be the first to report it formally and that this is likely an overly conservative assessment of novelty.

**Conditional analyses**. To further confirm the novelty of the allele associations found above, we used GCTA-COJO[61] to perform conditional analyses (--cojo-cond with --cojo-joint) on each significant phenotype-allele combinations found in the 2-field analysis.

**Replication analyses in the Estonian Biobank**. The Estonian Biobank is a population-based biobank with 212,955 participants in the current data freeze (2022v2). All biobank participants have signed a broad informed consent form and information on ICD codes is obtained via regular linking with the national Health Insurance Fund and other relevant databases, with majority of the electronic health records having been collected since 2004[62]. The diagnosis of autoimmune diseases were based on ICD-10 and ATC codes listed in Supplementary Data 18. Analyses were restricted to individuals with European ancestry.

HLA imputation of the Estonian Biobank genotype data was performed at the Broad Institute using the SNP2HLA tool[12]. The imputation was done for genotype data generated on the Global Screening Array v1. We performed separate additive logistic regression analysis with the imputed HLA alleles using SAIGE v1.0.7 with standard binary trait settings[63]. Logistic regression was carried out with adjustment for current age, age², sex and 10 PCs as covariates, analyzing only variants with a minimum minor allele count of 2.

**Synonymous variant association tests: comparisons of two field and 3 field HLA allele associations**. We compared association results of 3-field HLA alleles belonging to the same 2-field class to check if adding synonymous variants would change association results. To do this, we limited our analyses to 2-field HLA alleles for which there was at least one statistically significant 3-field HLA allele for any given phenotype ($p < 5 \times 10^{-8}/11$). We then examined four scenarios: 1) In cases with more than one statistically significant 3-field allele, we compared beta estimates of all alleles to the one with the lowest p-value using the t-test for unequal variances in R (Welch's t-test using the beta and standard error from Regenie in the HLA association studies above). 2) In cases with only one statistically significant 3-field allele and one or more statistically non-significant 3-field alleles, we also directly compared the beta estimate of the significant 3-field allele to the beta estimate of each of the other alleles using Welch's t-test. 3) In cases with only one significant 3-field allele and multiple non-significant 3-field alleles, we collapsed all non-significant 3-field alleles into a single allele. We then performed an association test of carrying this collapsed allele using the same covariates as our association tests above and compared the beta from that collapsed allele to the beta from the significant 3-field allele using Welch's t-test. For example, if the *HLA-A*01:01:01 allele was significant for a phenotype, but the *HLA-A*01:01:02 and *HLA-A*01:01:03 were not. We collapsed the latter two alleles into one and obtained a score of 0, 1, or 2 for each participant if they had none of these alleles, any one of the two, or any two of them, respectively.

This score was then used as our regressor. Note that this is equivalent to performing a gene-based burden test at any given HLA gene (here *HLA-A*), using only the count of statistically non-significant alleles (here *HLA-A*01:01:02 and *HLA-A*01:01:03) as the burden measurement to use as a regressor. This is precisely the way it was coded in Regenie to perform the analyses across HLA genes (either "--build-mask sum" or "--build-mask comphet" options). These burden tests were performed again separately for each ancestry and meta-analyzed as above. 4) Lastly, if there were multiple statistically significant 3-field alleles, but there remained non-significant 3-field alleles too, we also compared the non-significant 3-field alleles to the most significant 3-field alleles as per situation 2 or 3 above, depending on how many non-significant 3-field alleles there were. To determine if Welch's t-test was statistically significant, we used a Bonferroni correction for the number of Welch's t-tests divided by the number of tests performed in this section (i.e., $p < 0.05/307$).

**Canonical correlation analysis**. We used CCA[64] to find the total fraction of sequenced HLA alleles variance accounted for by the imputed HLA alleles. We assigned a value of 0, 1, or 2 to each allele (imputed and sequenced) based on whether it was absent, heterozygous, or homozygous, respectively. We then used the resulting two matrices as input to the yacca CCA R package[65], and obtained the total canonical redundancy[66] for sequenced HLA alleles (i.e., how much the imputed alleles were able to explain the sequenced alleles). This was done at multiple levels of sequenced allele frequencies: $> 0.01\%$, $> 0.1\%$, $> 1\%$, $> 5\%$, $> 10\%$, and $> 20\%$.

**Polygenic risk score**. We used polygenic risk scores to determine if the additional precision obtained from HLA sequencing at 3-field resolution would improve disease prediction performance. We first used the GWAS Catalog[67] to find GWAS summary statistics for our 11 phenotypes. We limited our search to studies with complete summary statistics, excluding those which only shared the most significant associations. We found complete summary statistics for 8 of those phenotypes: asthma[26], coeliac disease[68], type I diabetes mellitus[69], multiple sclerosis/demyelinating disease[70], psoriasis[33], rheumatoid arthritis[71], and ulcerative colitis[72]. Unfortunately, all found GWAS were on participants of European genetic ancestry, except for rheumatoid arthritis, which also contained East Asian ancestry participants (34.5% of the 103,638 participants in the GWAS). However, we used the entire UKB cohort for our polygenic risk score training and testing (regardless of ancestry assignment).

We computed a polygenic score from those summary statistics using the LDpred software[73] with the European HapMap[74] precompiled reference panel obtained from the LDpred developers (i.e. 1,054,330 variants). We used LDpred's genomic best linear unbiased predictor method (LDpred-inf, i.e., snp_ldpred2_inf in R). This was done in two ways: 1) using the GWAS summary statistics genome-wide, and 2) after removal of the chromosome 6 MHC region $+/-500$ kbp (i.e., GRCh37: 27,977,797 to 33,948,354; GRCh38: 28,010,120 to 33,980,577). Two LDpred scores were then assigned to each participant in the UKB (with and without the HLA region).

We then randomly split the participant set into a training set and a testing set at an 80/20 ratio and trained an XGBoost[75] random forest binary classifier using age, sex, the first 10 PCs (those projected on the 1000 G reference), and either of the following three sets of variables: 1) using only the LDpred score (with the HLA region), 2) using the LDpred score without the HLA region, and the imputed HLA alleles and 3) using the

LDpred score without the HLA region, and the sequenced HLA alleles at the 3-field resolution. The HLA alleles were assigned a value of 0, 1, or 2, as described in the CCA section above. The log loss was used with 5-fold cross-validation on the training set. We used a Bayesian optimization algorithm to tune the following XGBoost hyperparameters: max_depth, min_child_weight, eta, gamma, subsample, colsample_bytree, and max_delta_step. After training, we tested our 3 risk scores in the testing set and compared them using the AUC of the receiver operator characteristic curve. XGboost model training and testing was done on R.

**Protein normal mode analysis and contact map**. To study the interaction between the HLA-DQA1 and HLA-DQB2 hetero-dimer, we used the PDB file from the 5KSV entry of the Protein Data Bank[76,77], which gives the crystal structure of the HLA-DQ2.5 heterodimer (proteins of the *HLA-DQA1\*05:01* and *HLA-DQB1\*02:01*, with part of its CD74 ligand). Normal mode analysis was done using the C-alpha model with default options with the bio3d package[78] (v2.3-0) on R. Contact map was also done using the bio3d package.

**Reporting summary**. Further information on research design is available in the Nature Portfolio Reporting Summary linked to this article.

## Data availability
The primary data used for all analyses (i.e., WES CRAM files) is available through the UK Biobank DNAnexus research analysis platform. All summary statistics can be found in the supplements. All HLA-HD allele calls and their associated quality control metrics have been returned to the UK Biobank for sharing with authorized researchers. Source data underlying Fig. 1 are available in Supplementary Data 3, 4. Source data underlying Fig. 3 are available in Supplementary Data 20. Source data underlying Fig. 4 are available in Supplementary Data 24. Source data underlying Fig. 5 are available in Supplementary Data 26 (with exceptions, see below). For Fig. 6, the source data for the proteins is accessible publicly, and references are provided in the manuscript methods. The rest of the source data for Fig. 6 is available in Supplementary Data 20. For figures pertaining to UK Biobank participant-level data (e.g., Fig. 2, parts of Fig. 5, and Supplementary Fig. 1), primary data had been returned to the UK Biobank for sharing with authorized researchers as explained above.

## Code availability
All code is available on https://github.com/DrGBL/HLA_UKB[79].

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

## Acknowledgements

The Richards group is supported by the Canadian Institutes of Health Research (CIHR), the Lady Davis Institute of the Jewish General Hospital, the Canadian Foundation for Innovation, the NIH, Cancer Research UK, and FRQS. The Richards research group is supported by the Canadian Institutes of Health Research (CIHR: 365825; 409511, 100558, 169303), the McGill Interdisciplinary Initiative in Infection and Immunity (MI4), the Lady Davis Institute of the Jewish General Hospital, the Jewish General Hospital Foundation, the Canadian Foundation for Innovation, the NIH Foundation, Cancer Research UK, Genome Québec, the Public Health Agency of Canada, McGill University, Cancer Research UK [grant number C18281/A29019] and the Fonds de Recherche Québec Santé (FRQS). JBR is supported by an FRQS Mérite Clinical Research Scholarship. TwinsUK is funded by the Welcome Trust, Medical Research Council, European Union, the National Institute for Health Research (NIHR)-funded BioResource, Clinical Research Facility and Biomedical Research Centre based at Guy's and St Thomas' NHS Foundation Trust in partnership with King's College London. GBL received a scholarship from the FRQS and the CIHR. These funding agencies had no role in the design, implementation or interpretation of this study. The work of the Estonian Genome Center, University of Tartu was funded by the European Union through Horizon 2020 research and innovation program under grants no. 810645 and 894987, through the European Regional Development Fund projects GENTRANSMED (2014-2020.4.01.15-0012), MOBEC008 and Estonian Research Council Grant PRG1291. We want to acknowledge the participants of the Estonian Biobank and the UK Biobank for their contributions. The Estonian Biobank analyses were partially carried in the High Performance Computing Center, University of Tartu.

## Author contributions

G.B.L. and J.B.R. conceived the study. G.B.L. designed experiments with a contribution from J.F. and T.N.; G.B.L. analyzed the data with a contribution from T.L., Y.C., M.H., S.Y., Y.I., K.Y.H.L., C.Y.S., J.D.S.W., S.Z., D.T., and J.B.R.; G.B.L. wrote the manuscript. V.F. and J.B.R. contributed funding and resources. E.A., A.M., L.M., R.M., M.N., T.E., G.H., and Est.B.B. provided independent replication. All authors proof-read and approved the manuscript.

## Competing interests

The authors declare the following competing interests: JBR's institution has received investigator-initiated grant funding from Eli Lilly, GlaxoSmithKline and Biogen for projects unrelated to this research. He is the CEO of 5 Prime Sciences Inc (www.5primesciences.com). J.F., T.L., and V.F. are employees of 5 Prime Sciences Inc. T.N. has received a speaking fee from Boehringer Ingelheim for talks unrelated to this research. The other authors declare no competing interests.

## Ethics approval

The UKB was approved by the Northwest Multi-centre Research Ethics (approval number: 11/NW/0382). The activities of the Estonian Biobank are regulated by the Human Genes Research Act, which was adopted in 2000 specifically for the operations of Estonian Biobank. Individual level data analysis in Estonian Biobank was carried out under ethical approval 1.1-12/624 from the Estonian Committee on Bioethics and Human Research (Estonian Ministry of Social Affairs), using data according to release application 6-7/GI/8592 from the Estonian Biobank. Informed consent was obtained from all participants.

## Additional information

## Estonian Biobank Research Team

Erik Abner 8, Andres Metspalu 8, Lili Milani 8, Reedik Mägi 8, Mari Nelis8, Georgi Hudjashov8 & Tõnu Esko8

