## [Peer Review File · Communications Biology]

Reviewers' comments:

Reviewer #1 (Remarks to the Author):

This study focuses on the complexly polymorphic HLA region of the genome and the amino-acid level association of HLA alleles with diseases. Using the UK Biobank, the authors selected eleven autoimmune diseases and demonstrated the effectiveness of HLA allele calling from whole-exome sequencing data using the HLA-HD algorithm.

The manuscript was sent to me as a reviewer in the context of a prior review and point-by-point responses to that review. I found the responses from the authors to be largely convincing and appropriate; however, I was struck by the fact that the authors did not attempt to actually do any of the suggested analyses from the prior reviewers. While the authors might not see the value of these analyses, two of them will improve the manuscript and I would like to see the results included with a revised manuscript:

- 1) Adjusting the thresholds for quality control of the imputed data and reporting the resulting concordance.
 - a. While I agree that the major problem lies in less comprehensive imputation panels, this is testable and it would be of value to investigators actually planning to use exome-based HLA imputation to understand the improvement (and what is lost) with tightened thresholding.
- 2) Use permutation testing to confirm the association with rare HLA alleles.
 - a. I agree that they used rigorous multiple testing corrections that would be convincing for alleles with frequencies >1-2%, but the issue is that the allele frequencies are so low that type I errors can survive strict Bonferroni and p-value thresholding methods.
- 3) Include a discussion of what imputation panels were used and the other options that will further improve future imputation (e.g. TOPMED, All of Us)

Additional comments for this manuscript from my perspective focus on the polygenic risk score section. The authors demonstrate that imputed HLA types were as effective as whole genome-sequence based tag-variants for the diseases studied. This is not surprising at all given: 1) they used data that informed the association to calculate the PRS (i.e. the training and discovery set were not independent and 2) the HLA-allele did not improve what tag genetic variants provided to the scores. While the authors concede this point, they go on to discuss the improvement when HLA-alleles are added to a score devoid of tag variants from the region. Additionally, the sentence: "This did not lead to large differences in AUC except for coeliac disease (AUC decreased from 0.84 to 0.81, absolute difference: -0.03, 95% CI: -0.06 to -0.0004)." does not make a strong case for a "large difference". I think that the authors should the final paragraph of the PRS section as follows (I apologize that tracked changes was not available for me – the removal of the last sentence was intentional:

In this study, HLA imputation was performed using the same imprecise algorithm in both the training and the test set. In real-world applications, one would use a PRS developed with the HLA alleles from one imputation algorithm in a separate population, probably using a different HLA typing method (e.g. another imputation software). To mimic this scenario, we used the XGBoost weights from our PRS developed with imputed HLA alleles, and we used WES-based HLA alleles for the testing set's input features. This did not lead to large differences in AUC. HLA imputation provided a small improvement except for coeliac disease (AUC decreased from 0.84 to 0.81, absolute difference: -0.03, 95% CI: -0.06 to -0.0004). We conclude that HLA allele imputation may be useful for PRS, but most phenotypes will not benefit beyond the inclusion of the tag variants in the HLA region of the genome. Phenotypes that are more influenced by the HLA are more likely to benefit from this gain in precision.

Reviewer #2 (Remarks to the Author):

Summary:

The authors generally responded to my comments fairly. However, the authors did not add any analyses but only stated several comments and modified the text. Some of them satisfied me, but others did not make sense at all. The authors need to be more open to modifying their strategy based on the reviewer's comments, which should eventually improve the manuscript quality. Please see below.

Major comments:

1, Regarding my original comment 2-1:

Type 1 errors can be from unstable statistics (not well-calibrated P values) and insufficient multiple testing correction.

- They claimed that their strategy is valid since they used a Firth test implemented in Regenie, a method whose validity was previously discussed. This is partially satisfactory, but the validity was not shown in their specific analyses (ie. HLA-oriented analysis). I am not yet convinced that their analysis is free from inflated statistics. They need to address this point. They can conduct permutation analyses (e.g., they shuffle the case-control label and confirm they do not observe low p-values in this null situation). Or, if the permutation is computationally too expensive, they could alternatively provide the same analysis results for non-autoimmune diseases as a negative control, which should not have low P values.

- I agree that the authors used a stringent P-value threshold to adjust for multiple testing.

2, Regarding my original comment 2-3:

"The associations within the MHC locus do not necessarily support the causality of the HLA alleles.

There are many immunologically important genes at this locus. They need to compare the SNP associations at this locus and confirm that the HLA alleles have the top association."

This comment was based on the previous studies using this strategy (PMID: 26029868; 22286218).

The authors responded that they did not mean to conduct fine mapping ("we have been extremely careful to not infer any causality to our results"). I disagree with their opinions. The concept of causality is not only for fine mapping. The very basic interpretation of the genetic association study is that the associated allele (or the alleles in its tight LD) is the causal allele. If they claim that neither of the associated allele or the alleles in its tight LD is the causal allele, their results are really nonsense.

"As for comparing the HLA alleles with HLA SNPs, every single one of our GWAS was chosen because it had a lead significant HLA SNP finding in previous papers. This is therefore work that has already been reported. Further, given the complex LD in the HLA, the appropriate approach is not to confirm HLA allele findings using HLA SNPs association, but rather to use HLA allele fine-mapping (like our analyses) to better inform the often-cryptic HLA SNP association results. Indeed, all published GWAS that reported a lead SNP in the HLA locus have usually not attempted further SNP-based analyses, but rather "switched" to HLA allele analyses. Hence, performing SNP-based analyses at this point would not add any additional information to our results but would only serve to confuse readers more due to the complex linkage disequilibrium in the HLA."

I also disagree with this statement.

Yes, it is true that, in a standard GWAS, we should not discuss associations at the HLA locus due to the complex LD structure. However, this does not mean that, in a HLA allele oriented analysis (like in this manuscript) does not have to analyze SNPs at this locus. We need to jointly analyze HLA alleles and SNPs in the HLA region. Otherwise, we may falsely claim the HLA causality when non-HLA genes in the HLA locus are actually causal.

In addition, they claimed that "every single one of our GWAS was chosen because it had a lead significant HLA SNP finding in previous papers". Previously reported "lead significant HLA SNP finding" does not guarantee that ALL associations at the HLA locus only come from HLA alleles. They found

multiple INDEPENDENT associations at this locus. Do they believe ALL of them only comes from HLA alleles?

Moreover, how do they interpret the biological mechanisms of synonymous variants they reported in "Effect of synonymous variants"? Since such variants do not affect amino acid sequence, the mechanism could be driven by eQTL, as reported in a previous study (PMID: 32066938). If this is the case, we need to discuss SNPs in the regulatory regions; most of them are not HLA alleles.

3, Regarding my original comment 2-4:

"Given that most UK Biobank participants were assigned European genetic ancestry (>95%), we did not feel comfortable with a formal comparison of ancestry specific effect, as these would probably require a more diverse sample. For the interested readers, analyses results for each ancestry separately were also provided in the supplements."

I disagree with their opinion. The ancestral difference in genetic associations is a critical topic in current genetics. Also, they explained this study as "a multi-ancestry analysis" in the title of their manuscript. Therefore, it is a reasonable expectation for the readers to see the discussion on ancestry. As they stated, "this is likely because if an allele is causal, then it probably has the same biological effect across ancestries, and if it's not causal, then its association with a trait is simply biased by linkage disequilibrium", the trans-ancestry comparison can suggest how well their results capture the causal signals, and hence it is very useful to understand the relevance of their results.

4, Regarding my original comment 3:

My original comment: "Regarding the Fig 1B and 1E: They showed the "number of 3-field HLA alleles per genetic ancestry, divided by the number of participants in each ancestry." Is it a fair comparison? They need to consider the "saturation" of the observed signals. I believe, the EUR stats have an unfair downward bias in this analysis. At a certain point, the increase in the sample size has no more benefit for the detection of the new alleles. They need to consider a down-sampling test or related tests for this discussion."

They claimed that "the point of this figure is purely descriptive, and there was no attempt done at statistical testing."

I totally disagree with their idea. This is too misleading. In the paragraph, they apparently claim the ancestry differences. For example, they stated, "the finding that the AMR participants have a larger number of HLA alleles is consistent with other studies and reference panels which showed that native American populations have a high number of HLA alleles absent in other populations". Do they think, if it is not statistical testing, they can allow a very misleading index in the main results?

Reviewer #1 (Remarks to the Author):

This study focuses on the complexly polymorphic HLA region of the genome and the amino-acid level association of HLA alleles with diseases. Using the UK Biobank, the authors selected eleven autoimmune diseases and demonstrated the effectiveness of HLA allele calling from whole-exome sequencing data using the HLA-HD algorithm.

The manuscript was sent to me as a reviewer in the context of a prior review and point-by-point responses to that review. I found the responses from the authors to be largely convincing and appropriate; however, I was struck by the fact that the authors did not attempt to actually do any of the suggested analyses from the prior reviewers. While the authors might not see the value of these analyses, two of them will improve the manuscript and I would like to see the results included with a revised manuscript:

- 1) Adjusting the thresholds for quality control of the imputed data and reporting the resulting concordance.
 - a. While I agree that the major problem lies in less comprehensive imputation panels, this is testable and it would be of value to investigators actually planning to use exome-based HLA imputation to understand the improvement (and what is lost) with tightened thresholding.

This was done as suggested. We used two additional allele dosage thresholds: a more liberal one of 0.66 (2/3) and a stricter one of 0.9. The liberal threshold was chosen to be the lowest dosage threshold ensuring that only two alleles would be imputed per gene per participants. Otherwise, it could become impossible to differentiate between heterozygous alleles with low dosage and homozygous alleles. This was done for each ancestry and for each gene, and is available in the new supplementary tables ST6-ST11.

As a short summary of these results, we expected that by increasing the threshold we would decrease the number of imputed alleles, and would decrease concordance. Similarly, concordance could only improve by decreasing the threshold, which would increase the number of alleles being summed in the concordance calculation. Reassuringly, the difference was generally small. This is coherent with the original UK Biobank paper that computed those imputed alleles, and which did not see large differences in association studies with or without dosage thresholds (PMID: 30305743).

As an example, we show here a summary of the tables found in ST6 for the entire cohort:

Result	Class I			Class II							
	HLA-A	HLA-B	HLA-C	HLA-DPA1	HLA-DPB1	HLA-DQA1	HLA-DQB1	HLA-DRB1	HLA-DRB3	HLA-DRB4	HLA-DRB5
Usual threshold											
Concordance	0.904	0.895	0.911	0.968	0.861	0.472	0.665	0.768	0.705	0.647	0.804
Adjusted concordance	0.97	0.946	0.986	0.997	0.909	0.815	0.97	0.925	0.947	0.794	0.935
Liberal threshold											
Concordance	0.909	0.906	0.913	0.969	0.89	0.473	0.672	0.783	0.704	0.639	0.805
Adjusted concordance	0.976	0.957	0.989	0.998	0.939	0.817	0.98	0.943	0.95	0.81	0.937
Strict threshold											
Concordance	0.894	0.881	0.907	0.966	0.804	0.47	0.623	0.747	0.705	0.652	0.804
Adjusted concordance	0.959	0.932	0.983	0.995	0.848	0.811	0.908	0.899	0.938	0.769	0.931

We have also added the following text in the manuscript:

“Finally, using different allele dosing QC threshold for the imputed alleles only had a mild effect on concordance results, with an average decrease in adjusted concordance of 0.93 percentage point (range: 0.10 to 3.0) when using liberal dosage thresholds (see Methods), and an average increase in adjusted concordance of 2.01 percentage point (range 0.2 to 6.2) when using a stricter threshold (see supplementary tables ST6-11 for full comparisons).”

2) Use permutation testing to confirm the association with rare HLA alleles.

a. I agree that they used rigorous multiple testing corrections that would be convincing for alleles with frequencies >1-2%, but the issue is that the allele frequencies are so low that type I errors can survive strict Bonferroni and p-value thresholding methods.

We agree that low allele frequency variants can survive Bonferroni correction in the context of usual logistic regression due to a large imbalance in the case-control 2x2 table, which makes it less likely that the test statistic will reach the asymptotic chi-square null distribution. Firth regression alleviates this problem by applying a penalty towards the null hypothesis inversely proportional to the sample size (in this case to the allele frequency). The mathematical details and examples of application in genetics and in other fields has been published multiple times. We also note that the allele with the lowest frequency that reached our p-value threshold had a frequency of 0.059%, which still represents an allele count of more than 400 in the UK Biobank.

Nevertheless, since both reviewers expressed concerns, we used the second reviewer’s suggestion to use “negative control phenotypes”. That is, if using a phenotype that is not supposed to be associated with the HLA still results in HLA associations using our method, then, clearly, we didn’t adjust our Type 1 error appropriately. Doing permutation tests for Firth regression on a cohort the size of the UK Biobank would be computationally impractical.

Hence, we performed the same analyses using osteoporosis, atrial fibrillation, and non-insulin dependent diabetes mellitus. These are not expected to result in any HLA association, though we expect smaller p-values in non-insulin dependent diabetes mellitus due to its frequent “contamination” with type 1 diabetes mellitus in registry-based cohorts like the UK Biobank. This is exactly what we observed, as not a single association passed our p-value threshold of $5e-8/11$, and non-insulin dependent diabetes showed the smallest p-values.

We have added the following in our methods to explain the above:

“Given the use of rare alleles which may lead to unstable p-values and an increase in type 1 error, we also used negative controls phenotypes (osteoporosis, atrial fibrillation, non-insulin dependent diabetes mellitus) to ensure that this threshold and the Firth regression was enough to control the type 1 error rate. These phenotypes are expected to have no strong HLA association, except for non-insulin dependent diabetes mellitus which is often “contaminated” by Type 1 diabetes mellitus cases in large registries. As expected, none of these phenotypes showed significant HLA associations with our HLA alleles at $p < 5 \times 10^{-8/11}$ (see supplementary figure SF11 and supplementary table ST28).”

We also reproduce here the qq-Plots from the 3-field analyses to show how type 1 error rate was well controlled. Results for the 2-field analyses were the same. They can be found in supplement SF11 too. Full

summary statistics are also available in the supplementary tables (ST28). In the figures below, the horizontal dashed bars represent our p-value threshold ($p < 5e-8/11$):

3) Include a discussion of what imputation panels were used and the other options that will further improve future imputation (e.g. TOPMED, All of Us)

No imputation panel was used to obtain the new HLA alleles; we used whole-exome sequencing CRAM files to retrieve HLA alleles directly (after converting them to fastq). For HLA imputation, results were released by the UK Biobank and obtained using HLA:IMP*2. As previously discussed, this is not something that we did ourselves, but rather data that the UK Biobank has made available to researchers (see here: <https://biobank.ndph.ox.ac.uk/showcase/field.cgi?id=22182>).

The only time single variants imputation was used was when assigning genetic ancestry and in the first step of the Regenie software. In those situations, we also used the UK Biobank's imputed variants. This was described in detail in the UK Biobank's main paper (PMID: 30305743), and we have referred readers to it. This data is also made available to researchers by the UK Biobank directly (see here: <https://biobank.ndph.ox.ac.uk/showcase/field.cgi?id=21008>).

Additional comments for this manuscript from my perspective focus on the polygenic risk score section. The authors demonstrate that imputed HLA types were as effective as whole genome-sequence based tag-variants for the diseases studied. This is not surprising at all given: 1) they used data that informed the association to calculate the PRS (i.e. the training and discovery set were not independent and 2) the HLA-allele did not improve what tag genetic variants provided to the scores. While the authors concede this point, they go on to discuss the improvement when HLA-alleles are added to a score devoid of tag variants from the region. Additionally, the sentence: "This did not lead to large differences in AUC except for coeliac disease (AUC decreased from 0.84 to 0.81, absolute difference: -0.03, 95% CI: -0.06 to -0.0004)." does not make a strong case for a "large difference". I think that the authors should the final paragraph of the PRS section as follows (I apologize that tracked changes was not available for me – the removal of the last sentence was intentional:

In this study, HLA imputation was performed using the same imprecise algorithm in both the training and the test set. In real-world applications, one would use a PRS developed with the HLA alleles from one imputation algorithm in a separate population, probably using a different HLA typing method (e.g. another imputation software). To mimic this scenario, we used the XGBoost weights from our PRS developed with imputed HLA alleles, and we used WES-based HLA alleles for the testing set's input features. This did not lead to large differences in AUC. HLA imputation provided a small improvement except for coeliac disease (AUC decreased from 0.84 to 0.81, absolute difference: -0.03, 95% CI: -0.06 to -0.0004). We conclude that HLA allele imputation may be useful for PRS, but most phenotypes will not benefit beyond the inclusion of the tag variants in the HLA region of the genome. Phenotypes that are more influenced by the HLA are more likely to benefit from this gain in precision.

We have amended the paragraph exactly as suggested by the reviewer. For ease, we note that other than the removal of the end of the original paragraph, the main change is the addition of the sentence "We conclude that HLA allele imputation may be useful for PRS, but most phenotypes will not benefit beyond the inclusion of the tag variants in the HLA region of the genome". We of course agree with this interpretation of our results.

Reviewer #2 (Remarks to the Author):

Summary:

The authors generally responded to my comments fairly. However, the authors did not add any analyses but only stated several comments and modified the text. Some of them satisfied me, but others did not make sense at all.

The authors need to be more open to modifying their strategy based on the reviewer's comments, which should eventually improve the manuscript quality. Please see below.

Major comments:

1, Regarding my original comment 2-1:

Type 1 errors can be from unstable statistics (not well-calibrated P values) and insufficient multiple testing correction.

- They claimed that their strategy is valid since they used a Firth test implemented in Regenie, a method whose validity was previously discussed. This is partially satisfactory, but the validity was not shown in their specific analyses (ie. HLA-oriented analysis). I am not yet convinced that their analysis is free from inflated statistics. They need to address this point. They can conduct permutation analyses (e.g., they shuffle the case-control label and confirm they do not observe low p-values in this null situation). Or, if the permutation is computationally too expensive, they could alternatively provide the same analysis results for non-autoimmune diseases as a negative control, which should not have low P values.

We agree that low allele frequency variants can survive Bonferroni correction in the context of usual logistic regression due to a large imbalance in the case-control 2x2 table, which makes it less likely that the test statistic will reach the asymptotic chi-square null distribution. Firth regression alleviates this problem by applying a penalty towards the null hypothesis inversely proportional to the sample size (in this case to the allele frequency). The mathematical details and examples of application in genetics and in other fields has been published multiple times. We also note that the allele with the lowest frequency that reached our p-value threshold had a frequency of 0.059%, which still represents an allele count of more than 400 in the UK Biobank.

Nevertheless, since both reviewers expressed concerns, we used the second reviewer's suggestion to use "negative control phenotypes". That is, if using a phenotype that is not supposed to be associated with the HLA still results in HLA associations using our method, then, clearly, we didn't adjust our Type 1 error appropriately. Doing permutation tests for Firth regression on a cohort the size of the UK Biobank would be computationally impractical (and again, the goal of Regenie with Firth regression is to alleviate the need for these types of computations).

Hence, we performed the same analyses using osteoporosis, atrial fibrillation, and non-insulin dependent diabetes mellitus. These are not expected to result in any HLA association, though we expect smaller p-values in non-insulin dependent diabetes mellitus due to its frequent "contamination" by type 1 diabetes mellitus in registry-based cohorts like the UK Biobank. This is exactly what we observed, as not a single association passed our p-value threshold of $5e-8/11$, and non-insulin dependent diabetes showed the smallest p-values.

We have added the following in our methods to explain the above:

"Given the use of rare alleles which may lead to unstable p-values and an increase in type 1 error, we also used negative controls phenotypes (osteoporosis, atrial fibrillation, non-insulin dependent diabetes mellitus) to ensure that this threshold and the Firth regression was enough to control the type 1 error rate. These phenotypes are expected to have no strong HLA association, except for non-insulin dependent diabetes mellitus which is often "contaminated" by Type 1 diabetes mellitus cases in large registries. As expected, none of these phenotypes showed significant HLA associations with our HLA alleles at $p < 5 \times 10^{-8/11}$ (see supplementary figure SF11 and supplementary table ST28)."

We also reproduce here the qq-Plots from the 3-field analyses to show how type 1 error rate was well controlled. Results for the 2-field analyses were the same. They can be found in supplement SF11 too. Full summary statistics are also available in the supplementary tables (ST28). In the figures below, the horizontal dashed bars represent our p-value threshold ($p < 5e-8/11$):

- I agree that the authors used a stringent P-value threshold to adjust for multiple testing.

2, Regarding my original comment 2-3:

“The associations within the MHC locus do not necessarily support the causality of the HLA alleles. There are many immunologically important genes at this locus. They need to compare the SNP associations at this locus and confirm that the HLA alleles have the top association.”

This comment was based on the previous studies using this strategy (PMID: 26029868; 22286218).

We again agree that associations at the MHC locus with HLA alleles is not sufficient to prove causality, as we have highlighted in our response to the first round of reviews. Unfortunately, a simple comparison of p-values between genetic variants and HLA alleles would also not support causality any further, as even in SNP association studies the top association is not necessarily causal. The only additional information that this would provide is to show that there are SNPs that tag HLA alleles (and vice versa), which is information that has been previously published on multiple occasions in the context of genome-wide association studies for which summary statistics are already publicly available. In fact, as previously mentioned, we specifically chose these phenotypes because there were GWASs that found lead variants in the HLA region. This is therefore not novel data or result.

The authors responded that they did not mean to conduct fine mapping (“we have been extremely careful to not infer any causality to our results”). I disagree with their opinions. The concept of causality is not only for fine mapping. The very basic interpretation of the genetic association study is that the associated allele (or the alleles in its tight LD) is the causal allele. If they claim that neither of the associated allele or the alleles in its tight LD is the causal allele, their results are really nonsense.

We never claimed that there was an absence of causality, simply that just finding an association is not sufficient to infer causality at this association. Our original quote above was in the context of using stepwise regression (i.e. conditional analyses) to reinforce causality, which is not appropriate. We therefore agree with the reviewer, even if we have worded things slightly differently.

“As for comparing the HLA alleles with HLA SNPs, every single one of our GWAS was chosen because it had a lead significant HLA SNP finding in previous papers. This is therefore work that has already been reported. Further, given the complex LD in the HLA, the appropriate approach is not to confirm HLA allele findings using HLA SNPs association, but rather to use HLA allele fine-mapping (like our analyses) to better inform the often-cryptic HLA SNP association results. Indeed, all published GWAS that reported a lead SNP in the HLA locus have usually not attempted further SNP-based analyses, but rather “switched” to HLA allele analyses. Hence, performing SNP-based analyses at this point would not add any additional information to our results but would only serve to confuse readers more due to the complex linkage disequilibrium in the HLA.”

I also disagree with this statement.

Yes, it is true that, in a standard GWAS, we should not discuss associations at the HLA locus due to the complex LD structure. However, this does not mean that, in a HLA allele oriented analysis (like in this manuscript) does not have to analyze SNPs at this locus. We need to jointly analyze HLA alleles and SNPs in the HLA region. Otherwise, we may falsely claim the HLA causality when non-HLA genes in the HLA locus are actually causal.

As stated above, we never claimed that our HLA alleles were causal simply because we found an association. Again, we refer the reviewer and the readers to the already published publicly available summary statistics for each of these phenotypes for SNP level association.

Importantly, the two previous comments provide contradictory opinions of our work, and we believe there is a third, more appropriate way to interpret it. The former comment was concerned that we did not infer enough causality (“If they claim that neither of the associated allele or the alleles in its tight LD is the causal allele, their results are really nonsense”), while the latter was concerned that we inferred too much (“we may falsely claim the HLA causality when non-HLA genes in the HLA locus are actually causal”). We have always preferred a more cautious interpretation, which is the standard one for similar genetic association studies: barring bias such as population stratification (which we have done our best to control for), significant associations suggest either a causal (HLA) allele, or a causal tagging allele/SNP. We believe that our manuscript and choice of analyses reflect this interpretation.

In addition, they claimed that “every single one of our GWAS was chosen because it had a lead significant HLA SNP finding in previous papers”. Previously reported “lead significant HLA SNP finding” does not guarantee that ALL associations at the HLA locus only come from HLA alleles. They found multiple INDEPENDENT associations at this locus. Do they believe ALL of them only comes from HLA alleles?

As stated in the manuscript and the previous round of reviews, the choice of these phenotypes was not to “guarantee that all associations [...] only come from HLA alleles”. It was to make sure that the phenotypes would have enough power. For example, as stated in our response to the previous comments, we left out lupus because it had not shown to be a well-powered enough phenotype in the UK Biobank. Additionally, it provided us with a set of positive and negative control for HLA allele association by comparing with the UK Biobank imputed alleles, allowing us to verify the quality of our data. This was done as planned, and there was no ulterior causal interpretation to this choice.

As for the number of independent associations found, this is hardly surprising, since HLA alleles are by definition an amalgam of hundreds to thousands of SNPs, and that the HLA has evolved to be highly heterogeneous. Specifically, even if we assume full causality in our results (which we don't), finding multiple HLA alleles that are independently causal for a given phenotypes does not mean that the same SNP variant is driving causality, nor that all SNPs tagging the HLA alleles are causal. It just means that a combination of those SNPs is causal, and that the different combinations of SNPs are independent. This includes tagging SNP, including intronic SNPs that are collapsed into 2-fields and 3-fields (more on this below). We are confident that the reviewer does not mean to say that no HLA alleles are causal, and it would also be a fair reading of our manuscript to say that we do not mean that all HLA alleles are causal (and we highlight that there is no sentence in the manuscript to suggest such a statement).

Moreover, how do they interpret the biological mechanisms of synonymous variants they reported in “Effect of synonymous variants”? Since such variants do not affect amino acid sequence, the mechanism could be driven by eQTL, as reported in a previous study (PMID: 32066938). If this is the case, we need to discuss SNPs in the regulatory regions; most of them are not HLA alleles.

This is probably the most interesting and important question remaining. And in fact, we had acknowledged the lack of answers provided in our original manuscript, and specifically offered an initial path to answers:

“Third, HLA-HD does not provide 4-field resolution, which would be necessary to study non-coding variants. Given our findings on the non-negligible role of synonymous HLA variants in human disease, we expect that non-coding variants would also be important to study more thoroughly.”

This highlights two point:

- 1) All intronic variants are collapsed in the 3-field and 2-field analyses. Therefore, it is indeed possible that some of the synonymous variant signal actually tags an intronic variant. But this doesn't discredit our analyses, it in fact reinforces the key point that using HLA imputation method that stop at 2-field is likely a mistake if we want to better understand the HLA. Until 4-field becomes the norm, we show that 3-field provides clearer results than 2-fields.**
- 2) Synonymous variant can absolutely have non-negligible regulatory roles, affect gene expression, and even have protein altering roles (e.g. RNA splicing), but we will probably not be able to fully determine them on a large scale without going to 4-fields.**

We have therefore amended our original comment to the following:

“Third, HLA-HD does not provide 4-field resolution, which would be necessary to study non-coding variants, including those that may be tagging synonymous variant, explaining the signal we found when comparing 2-field to 3-field. Given our findings on potential the non-negligible role of synonymous HLA variants in human disease, we expect that non-coding variants would also be important to study more thoroughly.”

Note that this comment from the reviewer made us realize that we had not properly defined the second field of the HLA nomenclature in our manuscript. We had previously defined it as “protein-altering” for simplicity, when it should have been “HLA protein”. We have now fixed this.

3, Regarding my original comment 2-4:

“Given that most UK Biobank participants were assigned European genetic ancestry (>95%), we did not feel comfortable with a formal comparison of ancestry specific effect, as these would probably require a more diverse sample. For the interested readers, analyses results for each ancestry separately were also provided in the supplements.”

I disagree with their opinion. The ancestral difference in genetic associations is a critical topic in current genetics. Also, they explained this study as "a multi-ancestry analysis" in the title of their manuscript. Therefore, it is a reasonable expectation for the readers to see the discussion on ancestry.

As they stated, “this is likely because if an allele is causal, then it probably has the same biological effect across ancestries, and if it's not causal, then its association with a trait is simply biased by linkage disequilibrium”, the trans-ancestry comparison can suggest how well their results capture the causal signals, and hence it is very useful to understand the relevance of their results.

We have now added heterogeneity statistics for each allele and for each phenotype for which at least two genetic ancestries were meta-analyzed. These are available in the supplementary table ST20, for 2-field and 3-field analyses, as well as amino acids. As expected per our discussion above, we did not find strong evidence of heterogeneity for HLA alleles, with only one allele showing robust evidence of heterogeneity (using Bonferroni adjustment and qqPlot visual inspection): DOB*01:02:01 for type 1 diabetes mellitus. We added the following in the results:

*“Lastly, among phenotypes which were analyzed in more than one ancestry and could therefore be meta-analyzed, there were no strong signal of heterogeneity in HLA associations. Indeed, qq-Plots of heterogeneity p-values show that there were no heterogeneous effects using our genome-wide threshold ($p < 5 \times 10^{-8}/11$), and only one association with p-value lower than the Bonferroni threshold ($p < 0.05/13,576$) at DOB*01:02:01 for type 1 diabetes mellitus (see supplementary table **ST20** and supplementary figure **SF8**).”*

However, for amino acid associations, we observed significant heterogeneity for type 1 diabetes and autoimmune thyroid disorders (and asthma to a lesser extent), especially at class 2 HLA genes DOB, DRB1, DRB5, and DQB1. Given the number of heterogeneous associations found, a full description and explanation would likely deserve its own detailed analysis, outside the scope of this manuscript. Nevertheless, we offer a hypothesis that is similar to our previous explanation: if an amino acid is truly causal (or in high linkage disequilibrium with a genetic variant that is causal), then it should have a similar effect across ancestries, and if not, then we’re likely to see heterogeneity. Hence, since we do not expect all amino acid to be causally associated with the phenotypes, we believe this is the most likely explanation. More research will be needed to confirm, likely with methods outside the realm of pure genetics. We have added the following to our manuscript to summarize our results and hypothesis:

*“Lastly, in contrast to HLA alleles, we observed significant heterogeneity in amino acid associations for the autoimmune thyroid disorders, type I diabetes mellitus phenotypes, and asthma to a lesser extent (supplementary figure **SF8**). This was especially the case in class II HLA genes DOB, DRB1, DRB5, and DQB1 (supplementary table **ST20**). This potentially represents amino acid residues which are neither causal, nor highly correlated with causal genetic variants, but more research would be needed to confirm this hypothesis.”*

To better showcase these results, we reproduce here the qq-plots of heterogeneity statistics p-values for some of the analyses. The vertical dashed line is the Bonferroni corrected p-value threshold for the analysis. All plots for all phenotypes at 3-field, 2-field and for amino acids can be found in supplementary figure SF8.

4, Regarding my original comment 3:

My original comment: "Regarding the Fig 1B and 1E: They showed the "number of 3-field HLA alleles per genetic ancestry, divided by the number of participants in each ancestry." Is it a fair comparison? They need to consider the "saturation" of the observed signals. I believe, the EUR stats have an unfair downward bias in this analysis. At a

certain point, the increase in the sample size has no more benefit for the detection of the new alleles. They need to consider a down-sampling test or related tests for this discussion.”

They claimed that “the point of this figure is purely descriptive, and there was no attempt done at statistical testing.”

I totally disagree with their idea. This is too misleading. In the paragraph, they apparently claim the ancestry differences. For example, they stated, “the finding that the AMR participants have a larger number of HLA alleles is consistent with other studies and reference panels which showed that native American populations have a high number of HLA alleles absent in other populations”. Do they think, if it is not statistical testing, they can allow a very misleading index in the main results?

We now added the down sampling analysis as suggested in the previous review rounds. In summary, for the AFR, AMR, EUR, and SAS cohorts, for each classical HLA genes, we resampled 5,294 alleles from their respective set of alleles (without replacement). This number was chosen as twice the number of EAS participants, the smallest of the five ancestral cohorts. We only used classical HLA genes as the other ones did not have enough unique alleles in some cohort to provide stable results. This scheme was performed 10,000 times to provide intervals. As expected, the AMR cohort almost always had more unique alleles than other ancestries, except for DPA1, for which the AFR ancestry had more unique alleles. The following was added to the manuscript to explain the results:

“Lastly, to adjust for the large differences in participants of each ancestry which could saturate the number of alleles found in each group, and hence bias comparisons on number of alleles per ancestry groups, we down sampled 10,000 times each group to have the same sample size and number of HLA alleles as the smallest group (EAS). This analysis was only done for classical HLA genes, as they had enough unique alleles for the simulations to be stable. As expected, the AMR ancestry cohort had more expected unique alleles than other groups in all genes, except for HLA-DPA1 where the AFR cohort, had more (supplementary table ST2 and supplementary figure SF3).”

The results are also summarized in supplementary table ST2, and graphically in supplementary figure SF3, which we reproduce below. In the figure, the red dashed vertical lines represent the number of unique alleles in the EAS ancestry, and each color represents the number of unique alleles found in each of the 10,000 iterations for their respective ancestry, illustrated as histograms.

REVIEWERS' COMMENTS:

Reviewer #1 (Remarks to the Author):

The authors have presented direct and sufficient answers to this reviewer's queries.

Reviewer #2 (Remarks to the Author):

The authors generally addressed my comments. The "negative control phenotypes" analysis (SF11) makes sense and is a reasonable alternative to a permutation test. It looks very convincing, addressing my concerns about the type I error rate. I do not have further comments.